# TuneUp: A Training Strategy for Improving Generalization of Graph Neural Networks

## Abstract

Despite many advances in Graph Neural Networks (GNNs), their training strategies simply focus on minimizing a loss over nodes in a graph. However, such simplistic training strategies may be sub-optimal as they neglect that certain nodes are much harder to make accurate predictions on than others. Here we present TuneUp, a curriculum learning strategy for better training GNNs. Crucially, TuneUp trains a GNN in two stages. The first stage aims to produce a strong base GNN. Such base GNNs tend to perform well on head nodes (nodes with large degrees) but less so on tail nodes (nodes with small degrees). So, the second stage of TuneUp specifically focuses on improving prediction on tail nodes. Concretely, TuneUp synthesizes many additional supervised tail node data by dropping edges from head nodes and reusing the supervision on the original head nodes. TuneUp then minimizes the loss over the synthetic tail nodes to finetune the base GNN. TuneUp is a general training strategy that can be used with any GNN architecture and any loss, making TuneUp applicable to a wide range of prediction tasks. Extensive evaluation of TuneUp on five diverse GNN architectures, three types of prediction tasks, and both inductive and transductive settings shows that TuneUp significantly improves the performance of the base GNN on tail nodes, while often even improving the performance on head nodes, which together leads up to 58.5% relative improvement in GNN predictive performance. Moreover, TuneUp significantly outperforms its variants without the two-stage curriculum learning, existing graph data augmentation techniques, as well as other specialized methods for tail nodes.

## 1 Introduction

Graph Neural Networks (GNNs) are one of the most successful and widely used paradigms for representation learning on graphs, achieving state-of-the-art performance in a variety of prediction tasks, such as semi-supervised node classification (Kipf & Welling, 2017; Velickovic et al., 2018), link prediction (Hamilton et al., 2017; Kipf & Welling, 2016), and recommender systems (Ying et al., 2018; He et al., 2020). There has been a surge of work on improving GNN model architectures (Velickovic et al., 2018; Xu et al., 2019; 2018; Shi et al., 2020; Klicpera et al., 2019; Wu et al., 2019; Zhao & Akoglu, 2019; Li et al., 2019; Chen et al., 2020; Li et al., 2021) and task-specific losses (Kipf & Welling, 2016; Rendle et al., 2012; Verma et al., 2021; Huang et al., 2021). Despite all these advances, strategies for training a GNN on a given supervised loss remain largely simplistic. Existing work has focused on simply minimizing the given loss over nodes in a graph. While such a simplistic default strategy already gives a strong performance, the strategy may still be sub-optimal as it neglects that some nodes are much harder to make accurate predictions on than others. Consequently, a GNN trained with the default strategy may significantly under-perform on those hard nodes, resulting in overall sub-optimal predictive performance.

Here we present TuneUp to better train a GNN on a given supervised loss. The key motivation behind TuneUp is that GNNs tend to under-perform on tail nodes, *i.e.*, nodes with a small number of neighbors (Liu et al., 2021). In practice, performing well on tail nodes is important since they are prevalent in real-world scale-free graphs (Clauset et al., 2009) and newly-arriving cold-start nodes (Lika et al., 2014). To better train a GNN on those hard-to-predict tail nodes, the key idea of TuneUp is to use a curriculum learning strategy (Bengio et al., 2009); TuneUp first trains a GNN

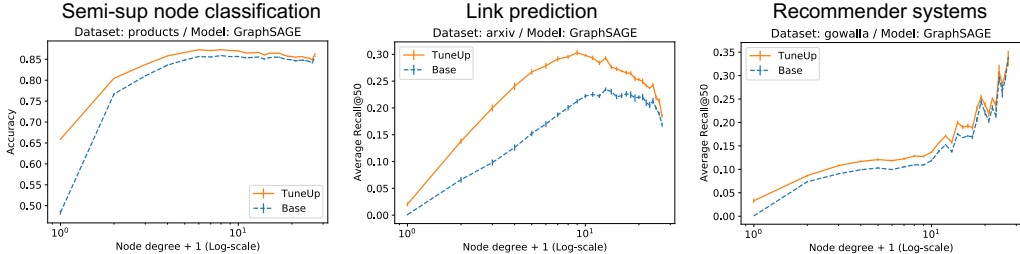

Figure 1: Degree-specific generalization performance of the base GNN and TUNEUP in the transductive setting. The $x$-axis represents the node degrees in the training graph, and the $y$-axis is the generalization performance averaged over nodes with the specific degrees. We see from the dotted blue curves that the base GNN tends to perform poorly on tail nodes, *i.e.*, nodes with small degrees. Our TUNEUP (denoted by the solid orange curves) improves or at least maintains the base GNN performance on almost all node degrees. The improvement is more significant on tail nodes.

to perform well on relatively easy head nodes, *i.e.*, nodes with a large number of neighbors. It then proceeds to improve the performance on the hard tail nodes.

Specifically, TUNEUP uses the two-stage strategy to train a GNN. In the first stage, TUNEUP employs the default training strategy, *i.e.*, simply minimizing the given supervised loss, to produce a strong base GNN to start with. The base GNN tends to perform well on head nodes, but poorly on tail nodes (see the dotted blue curves in Figure 1). To mitigate this issue, the second stage of TUNEUP focuses on improving the performance on the tail nodes. Specifically, TUNEUP synthesizes many additional tail node inputs by dropping edges from head nodes. TUNEUP then adds target supervision (*e.g.*, class labels for node classification, edges for link prediction) on the synthetic tail nodes by reusing the supervision on the original head nodes (before dropping edges). Finally, TUNEUP finetunes the base GNN by minimizing the loss over the increased supervised tail node data. The dedicated training on the synthetic tail nodes allows the resulting GNN to perform much better on the real tail nodes, while often even improving the performance on head nodes. TUNEUP is simple to implement on top of the default training pipeline of GNNs, as shown in Algorithm 1. Moreover, TUNEUP can be used to train any GNN model with any supervised loss, making it generally applicable to a broad range of node and edge-level prediction tasks.

We extensively evaluate TUNEUP on a wide range of settings. We consider five diverse GNN architectures, three types of key prediction tasks (semi-supervised node classification, link prediction, and recommender systems) with a total of eight datasets, as well as both transductive (*i.e.*, prediction on nodes seen during training) and inductive (*i.e.*, prediction on new nodes never seen during training) settings. For the inductive setting, we additionally consider the challenging cold-start scenario (*i.e.*, limited edge connectivity from new nodes) by randomly removing certain portions of edges from new nodes.

Across all settings, TUNEUP produces consistent improvement on the generalization performance of GNNs. In the transductive setting, TUNEUP significantly improves the performance of base GNNs on tail nodes, while oftentimes even improving the performance on head nodes (see Figure 1). Moreover, our ablation study shows that the two-stage curriculum training strategy of TUNEUP is critical and gives significantly improved performance over its variant strategy without curriculum learning. Finally, we extensively compare our TUNEUP against recent graph augmentation techniques (Rong et al., 2020; Liu et al., 2022) and specialized methods for tail nodes (Liu et al., 2021; Zheng et al., 2022; Zhang et al., 2022; Kang et al., 2022). Our TUNEUP outperforms all these methods in all settings, while being simpler and more general. Overall, our work demonstrates that training strategies can play an important role in improving generalization performance of GNNs.

## 2  GENERAL SETUP AND TUNEUP

TUNEUP is a curriculum learning strategy to train any GNN model with any supervised loss to solve node or edge-level prediction tasks over graphs. We first provide a general task setup for machine learning on graphs and review the default training strategy of GNNs to solve the task. We then present

TUNEUP, which adds a few simple components to the default training strategy. Finally, we discuss assumptions TUNEUP exploits to improve generalization performance of GNNs and why TUNEUP even improves the performance on head nodes.

## 2.1 GENERAL TASK SETUP

We are given a graph $G = (V, E)$, with a set of nodes $V$ and edges $E$ with potentially some features associated with them. GNN $F_\theta$, parameterized by $\theta$, takes the graph $G$ as input and makes prediction $\widehat{Y}$ for the task of interest. The loss function $L$ measures the discrepancy between the GNN's prediction $\widehat{Y}$ and the target supervision $Y$. In the default training, GNN parameter $\theta$ is learned to minimize the loss $L(\widehat{Y}, Y)$ using gradient descent. The setup is general to cover most graph machine learning scenarios. Below, we describe three representative scenarios under the general task setup, which we also consider in our experiments.

**Semi-supervised node classification**. The task is to predict class labels of unlabeled nodes given a small set of labeled nodes in a graph, which can be formalized as follows.

- **Graph** $G$: A graph with input node features.
- **Supervison** $Y$: Class labels of labeled nodes $V_{\text{labeled}} \subset V$.
- **GNN** $F_\theta$: A model that takes $G$ as input and predicts class probabilities over $V$.
- **Prediction** $\widehat{Y}$: The GNN's prediction over $V_{\text{labeled}}$.
- **Loss** $L$: Cross-entropy loss.

Since input node features are available, the GNN $F_\theta$ can make not only transductive predictions, *i.e.*, prediction over $V_{\text{unlabeled}} \equiv V \setminus V_{\text{labeled}}$, but also inductive predictions (Hamilton et al., 2017), *i.e.*, prediction over new nodes $V_{\text{new}}$ that are not in $V$ but connected to $V$ via new edges $E_{\text{new}}$.

**Link prediction**. The task is to predict new links in a graph given existing links. We consider the node-centric formulation (You et al., 2021): given a source node, predict target nodes that the source node is linked to.

- **Graph** $G$: A graph with input node features.
- **Supervison** $Y$: Whether node $s \in V$ is linked to node $t \in V$ in $G$ (positive) or not (negative).
- **GNN** $F_\theta$: A model that takes $G$ as input and predicts the score for a pair of nodes $(s, t) \in V \times V$. Specifically, the model generates embedding $\boldsymbol{z}_v$ for each node in $v \in V$ and uses an MLP over the concatenation of $\boldsymbol{z}_s$ and $\boldsymbol{z}_t$ to predict the score for the pair $(s, t)$ (He et al., 2017).
- **Prediction** $\widehat{Y}$: The GNN's predicted scores over $V \times V$.
- **Loss** $L$: The Bayesian Personalized Ranking (BPR) loss (Rendle et al., 2012), which encourages the predicted score for the positive pair $(s, t_{\text{pos}})$ to be higher than that for the negative pair $(s, t_{\text{neg}})$ for each source node $s \in V$.

As input node features are available, the GNN $F_\theta$ can naturally make inductive link prediction by generating node embeddings on a new graph with new nodes and edges.

**Recommender systems**. A recommender system can be modeled as a bipartite graph between user nodes $V_{\text{user}}$ and item nodes $V_{\text{item}}$, where edges represent user-item interactions. The task is essentially link prediction, *i.e.*, given a user node $u \in V_{\text{user}}$, predict a set of item nodes that $u$ is likely to interact with. In recommender systems, the most successful paradigm is collaborative filtering (Schafer et al., 2007), where shallow embeddings (learnable embeddings for each node) instead of input node features are used to achieve state-of-the-art performance (Wang et al., 2019; He et al., 2020). As input node features are not available in many public recommender system datasets anyway, we focus on the feature-less setting.

- **Graph** $G$: User-item bipartite graph *without* input node features.
- **Supervison** $Y$: Whether a user node $u$ has interacted with an item node $v$ in $G$ (positive) or not (negative).
- **GNN** $F_\theta$: A model that takes $G$ as input and predicts the score for a pair of nodes $(u, v) \in V_{\text{user}} \times V_{\text{item}}$. Following Wang et al. (2019), GNN parameter $\theta$ contains the input shallow embeddings in addition to the original message passing GNN parameter. To produce the score for the pair of nodes $(u, v)$, we generate the user and item embeddings, $\boldsymbol{z}_u$ and $\boldsymbol{z}_v$, and take the inner product $\boldsymbol{z}_u^\top \boldsymbol{z}_v$ to compute the score (Wang et al., 2019).
- **Prediction** $\widehat{Y}$: The GNN's predicted scores over $V_{\text{user}} \times V_{\text{item}}$.

---

**Algorithm 1** TUNEUP. Compared to the default training of a GNN (L2–5), TUNEUP introduces the two-stage training and only adds two components (L8 and L12) that are straightforward to implement.

---

**Given:** GNN $F_\theta$, graph $G$, loss $L$, supervision $Y$, DropEdge ratio $\alpha$.

1: **# First stage: Default training to obtain a base GNN.**
2: **while** $\theta$ not converged **do**
3:     Make prediction $\widehat{Y} = F_\theta(G)$
4:     Compute loss $L(\widehat{Y}, Y)$, compute gradient $\nabla_\theta L$, and update parameter $\theta$.
5: **end while**
6: **# Set up for the second stage.**
7: **if** task is semi-supervised node classification **then**
8:     Use $F_\theta$ to predict pseudo labels on non-isolated, unlabeled nodes. Add the pseudo labels into $Y$.
9: **end if**
10: **# Second stage: Fine-tuning the base GNN with increased tail supervision.**
11: **while** $\theta$ not converged **do**
12:     Synthesize tail nodes, *i.e.*, randomly drop $\alpha$ of edges: $G \xrightarrow{\text{DropEdge}} \widetilde{G}$.
13:     Make prediction $\widehat{Y} = F_\theta(\widetilde{G})$.
14:     Compute loss $L(\widehat{Y}, Y)$, compute gradient $\nabla_\theta L$, and update parameter $\theta$.
15: **end while**

---

- **Loss $L$:** The BPR loss (Rendle et al., 2012).

As we learn the shallow embedding for each node, it is non-trivial to make inductive predictions on new nodes. Therefore, we only consider the transductive setting for recommender systems.

## 2.2 DEFAULT GNN TRAINING

Given the graph $G$, supervision $Y$, GNN $F_\theta$, its prediction $\widehat{Y} = F_\theta(G)$, and the loss function $L(\widehat{Y}, Y)$, it is straightforward to train the GNN $F_\theta$ using gradient descent to minimize the loss. The default training procedure of a GNN is described in L2–5 of Algorithm1.

**Remark on mini-batch training**. In practice, the prediction $\widehat{Y}$ and the loss computation $L(\widehat{Y}, Y)$ can be made in a mini-batch manner for scalable training. For instance, in the case of semi-supervised node classification, we can predict and compute the loss on a mini-batch of labeled nodes (Hamilton et al., 2017; Zeng et al., 2020). In the case of link prediction and recommender systems, to compute the BPR loss, the score prediction only needs to be made over positive links and randomly sampled negative links. Moreover, the BPR loss can be computed in a mini-batch manner by subsampling the source nodes and keeping only one positive link per source node. These mini-batch training tricks are hidden in Algorithm 1 for simplicity, but should be implemented in practice. Our TUNEUP, which we explain next, is fully compatible with mini-batching training.

## 2.3 TUNEUP

We are ready to present TUNEUP that uses a two-stage curriculum learning strategy (Bengio et al., 2009) to better train a GNN. At high level, TUNEUP first trains a GNN to perform well on relatively easy head nodes and then proceeds to finetune the GNN to also perform well on hard tail nodes. Specifically, in the first stage (L2–5 in Algorithm 1), TUNEUP uses the default training of GNNs to obtain a strong base GNN model. The base GNN model tends to perform well on head nodes, but poorly on tail nodes. To remedy this issue, in the second training stage, TUNEUP finetunes the base GNN with increased supervision on tail nodes (L7–L15 in Algorithm 1). TUNEUP increases the supervised tail node data in two steps: (1) synthesizing additional tail node inputs and (2) adding target supervision on the synthetic tail nodes, which we detail below.

**(1) Synthesizing tail node inputs**. TUNEUP synthesizes many additional tail nodes by removing edges from the head nodes. Specifically, in this work, we directly adopt DropEdge (Rong et al., 2020) for simplicity, where a certain portion (given by hyperparameter $\alpha$) of edges are randomly removed from the original graph $G$ to obtain $\widetilde{G}$ (L12 in Algorithm 1). The resulting $\widetilde{G}$ contains more nodes with low degrees, *i.e.*, tail nodes, than the original graph $G$ does. Hence, the GNN sees more (synthetic) tail nodes as input during training. More advanced strategies to synthesize tail nodes (*e.g.*, dropping more ratio of edges from head nodes) are left for future work.

**(2) Adding supervision on the synthetic tail nodes**. After synthesizing the tail node inputs, TUNEUP adds target supervision (*e.g.*, class labels for node classification, edges for link prediction) on them so that the supervised loss can be computed over the synthetic tail nodes.

For link prediction tasks, TUNEUP directly reuses the original edges $E$ in $G$ (before dropping) as the target supervision on the synthetic tail nodes. To describe the effectiveness of the approach, suppose we have a node $v$ with six neighbors in the original training graph $G$. After dropping $\alpha = 0.5$ of edges in L12 of Algorithm 1, this node becomes a synthetic tail node $\widetilde{v}$ with three neighbors in $\widetilde{G}$. Nevertheless, in the loss computation of L14, TUNEUP still reuses *all original six edges* from $v$ in $G$ as target supervision on this synthetic node $\widetilde{v}$. Therefore, this synthetic tail node $\widetilde{v}$ has twice as much edge supervision as any degree-three real tail node (in the original graph $G$) has.

Similarly, for semi-supervised node classification, TUNEUP can also reuse the target labels of labeled nodes in $G$ for synthetic tail nodes in $\widetilde{G}$. Specifically, for a labeled node $v \in V_{\text{labeled}}$ with ground-truth class label $y_v$, TUNEUP can reuse $y_v$ for the corresponding synthetic tail node $\widetilde{v}$ in $\widetilde{G}$. However, in the semi-supervised setting, the number of labeled nodes $V_{\text{labeled}}$ is often very small, *e.g.*, 1%–5% of all nodes $V$, limiting the amount of target label supervision TUNEUP can reuse.

To resolve this issue, TUNEUP utilizes pseudo labels (Lee et al., 2013) in addition to the limited ground-truth labels on $V_{\text{labeled}}$. Specifically, TUNEUP applies the base GNN (obtained in the first training stage) over $G$ to predict pseudo labels on non-isolated (*i.e.*, positive-degree) nodes in $V_{\text{unlabeled}}$. In practice, the pseudo labels do not need to be directly predicted by the base GNN, *e.g.*, we can apply post-processing, such as label smoothing and C&S, to refine the pseudo labels. We leave the investigation to the future work.

TUNEUP then includes the pseudo labels as supervision $Y$ in the second stage (L8 in Algorithm 1). This would significantly increase the size of the supervision $Y$, *e.g.*, by a factor of $\approx$100 if only 1% of nodes are labeled. While the predicted pseudo labels are noisy in general, they are "best guesses" in the sense that the base GNN uses the *full graph information $G$* to predict the labels. In the second stage, TUNEUP essentially forces the base GNN to maintain its best guesses *given sparser graph $\widetilde{G}$ with limited neighborhood as input*. This in turn allows the resulting GNN to make more accurate prediction on *real* tail nodes with limited neighborhood in the original graph $G$.

### 2.4 ASSUMPTIONS TUNEUP EXPLOITS TO IMPROVE GENERALIZATION PERFORMANCE

As the no-free-lunch theorem suggests (Wolpert, 1996), improving generalization performance involves exploiting additional assumptions on real-world prediction tasks, which may not be satisfied by all possible tasks. Here we discuss three key assumptions TUNEUP exploits to improve generalization performance, which are (approximately) satisfied by many real-world tasks, including our experimented benchmark datasets across three different task types in Section 4.

**Tail nodes can be synthesized by dropping edges from head nodes**. This assumption holds for many real-world graph datasets, as head nodes often start off as tail nodes, *e.g.*, well-cited paper nodes are not cited at the beginning in a paper citation network, and warm users (users with many item interactions) start off as cold-start users in recommender systems.

**Target supervision on head nodes can be reused for synthetic tail nodes**. This assumption holds for tasks where prediction to be made on a given node is more or less a static property of the node. For instance, papers' subject areas in a paper citation network, products' categories in a product co-purchasing network, and users' taste in recommender systems stay (mostly) the same regardless of the number of edges we observe on the nodes.

**More edges benefit GNNs to make accurate predictions**. TUNEUP assumes that more edges are useful for GNNs to make accurate predictions, and tail nodes are harder to predict due to the lack of edges. This assumption is likely to hold for many tasks as GNNs can aggregate more neighboring information with more edges, and is experimentally verified in Figure 1.

### 2.5 WHY TUNEUP IMPROVES PERFORMANCE ON HEAD NODES

It is counter-intuitive that TUNEUP improves performance not only on tail nodes but also on head nodes, as seen in Figure 1. One reason may be that best-performing GNNs for node/edge-level tasks, including our experimented ones, use (roughly) *average-based* schemes to aggregate neighboring node features (Hamilton et al., 2017; Kipf & Welling, 2017; Wu et al., 2019; Velickovic et al.,

2018). With the average-based GNNs, node embeddings obtained on the sparsified graph $\widetilde{G}$ can be thought of as the *noisy version* of the node embeddings obtained on the full graph $G$. If the base GNN is finetuned to perform well on many realizations of the noisy embeddings (with different realizations of $\widetilde{G}$ in L12 of Algorithm 1), then the resulting GNN would most likely still perform well on the noise-free embeddings (computed over the full graph $G$). Moreover, training with the noisy embeddings can even improve the generalization performance on head nodes. We leave in-depth empirical/theoretical investigation for future work.

## 3 RELATED WORK

**Data augmentation for GNNs**. The second stage of TUNEUP can be regarded as data augmentation over graphs, on which there has been rich body of work (Ding et al., 2022). Some are specifically designed for semi-supervised node classification (Zhao et al., 2021; Feng et al., 2020; Verma et al., 2021), while others are designed for recommender systems (Verma et al., 2021) and graphs with input node features (Liu et al., 2022). Different from these methods, TUNEUP is generally applicable to any prediction tasks over nodes and edges. The general nature of TUNEUP also allows it to be combined with any of the task-specific data augmentation techniques. As a general graph augmentation technique, Kong et al. (2020) proposed FLAG, which adversarially perturbs input node features (Shafahi et al., 2019). This is complementary to TUNEUP, which perturbs the edge connectivity of the graph. DropEdge (Rong et al., 2020) randomly drops edges from graphs. It was originally developed to overcome the over-smoothing issue of GNNs (Li et al., 2018) in semi-supervised node classification. In contrast, in this work, we adopt DropEdge as a way to synthesize additional tail node inputs for a wide range of prediction tasks over graphs. Methodologically, TUNEUP is distinct from DropEdge in that it employs the two-stage curriculum learning strategy and uses pseudo labels to add supervision on the synthetic tail nodes, both of which are important to yield substantially better performance than the original DropEdge.

**Curriculum learning for GNNs**. A few works have explored curriculum learning for GNNs. Wang et al. (2021) developed a curriculum learning approach for graph classification, while our work focuses on node/edge-level prediction tasks. Ying et al. (2018) presented a curriculum learning for negative sampling in link prediction, and Li et al. (2022) developed a curriculum learning for tackling imbalanced class labels in node classification. TUNEUP is complementary to both of these approaches while being more broadly applicable to any node/edge-level prediction tasks.

**Pre-training GNNs**. Pre-training GNNs has attracted huge attention (Veličković et al., 2019; Hu et al., 2020b; Qiu et al., 2020; Hu et al., 2020c; You et al., 2020b). This line of work develops *task-agnostic* strategies to pre-train a GNN such that the resulting GNN can be finetuned with task-specific supervised losses to improve performance on diverse downstream tasks. Our work focuses on the downstream stage and presents a strategy for training a GNN on a *task-specific supervised loss*.

**Specialized methods for tail nodes**. Recently, many methods have been developed for improving generalization performance of GNNs on tail nodes (Liu et al., 2021; Zheng et al., 2022; Kang et al., 2022; Zhang et al., 2022). These methods require augmenting a GNN with tail-node-specific architectural components, while our work does not require any architectural modification and focuses purely on a strategy for *training a GNN* that performs well on *both tail and head nodes*.

## 4 EXPERIMENTS

Here we extensively evaluate TUNEUP under a wide range of settings. We consider five diverse GNN models and test them on the three prediction tasks described in Section 2.1 for three different predictive settings: transuctive, inductive, and cold-start inductive predictions.

### 4.1 EXPERIMENTAL SETTINGS

Here we describe our experimental settings and datasets for evaluating TUNEUP. We noticed that the standardized experimental protocols by Hu et al. (2020a); Wang et al. (2019) are not suitable for evaluating TUNEUP because (1) inductive prediction (cold-start) settings are not provided and (2) datasets are heavily pre-processed to eliminate tail nodes (*e.g.*, recommender system benchmarks are processsed with the 10-core algorithm to eliminate the cold-start users and items (Wang et al., 2019)), which is the focus of this work. We therefore take their original realistic graph datasets and split

them ourselves to create the realistic inductive (cold-start) prediction setting as well as the realistic transductive setting with tail nodes. The dataset statistics are summarized in Table 5 in Appendix. Below, we describe the split and the datasets for each task type.

**Semi-supervised node classification**. Given the entire nodes in the original dataset, we randomly selected 95% of the nodes and used their subgraph induced as the graph $G = (V, E)$ to train GNNs. The remaining 5% of the entire nodes, $V_{new}$, are used for inductive prediction. Within $V$, 10% and 2% of the nodes are used as labeled nodes $V_{labeled}$ for arxiv and products, respectively. A half of $V_{labeled}$ is used for computing the loss for supervised training, and another half is used as the transductive validation set for tuning hyper-parameters. For the evaluation metric, we used the standard classification accuracy. For the transductive performance, we report the accuracy on the unlabeled nodes $V_{unlabeled} \equiv V \setminus V_{labeled}$, while for the inductive performance, we report the accuracy on $V_{new}$. For the inductive prediction, we also consider the *cold-start* scenario, where certain portions (30%, 60%, and 90%) of edges are randomly removed from the new nodes. We used the following two datasets in our experiments.

- **arxiv** (Hu et al., 2020a): Given a paper citation network, the task is to predict the subject areas of the papers. Each paper has abstract words as its feature.
- **products** (Hu et al., 2020a): Given a product co-purchasing network, the task is to predict the categories of the products. Each product has the product description as its feature.

**Link prediction**. We follow the standard link prediction evaluation (Zhang & Chen, 2018; You et al., 2021) and randomly split the edges in the original graph into training and validation edges with the ratio of 50%/50%. We follow the same protocol as semi-supervised node classification to obtain nodes for transductive and inductive settings. For the evaluation metric, we used the recall@50 averaged over nodes (Wang et al., 2019), where the positive target nodes are scored among all negative nodes. For the transductive performance, we report the recall@50 computed over validation edges within $V$, while for inductive setting, we report the recall@50 over validation edges from $V_{new}$. For the inductive setting, we also consider the cold-start scenario. We used the following three datasets in our experiments.

- **flickr** (Zeng et al., 2020): Given an incomplete image-image common-property (*e.g.*, same geographic location, same gallery, comments by the same user, etc.) network, the task is to predict the new common-property links between images. Each image has its description has its feature.
- **ppi** (Chandak et al., 2022): Given an incomplete protein-protein interaction network, the task is to predict new interactions. Each protein feature is generated with ESM protein language model (Rives et al., 2021) applied to the protein sequence.
- **arxiv** (Hu et al., 2020a): Given an incomplete paper citation network, the task is to predict the additional citation links. Each paper has words in its abstract as its feature.

**Recommender systems**. For recommender systems, we notice that widely-used benchmark datasets are heavily processed to eliminate all tail nodes, *e.g.*, via the 10-core algorithm (Wang et al., 2019). For example, with the conventional 80%/20% train/validation split, the median training interactions per user is 17, 26, and 27 for gowalla, yelp2018, and amazon-book, respectively, which clearly do not reflect the realistic use case that involves many cold-start users and items (Lika et al., 2014). To reflect the realistic use case, we use the small training edge ratio on top of the existing benchmark datasets. Specifically, we randomly split the edges in the original graph into training and validation edges with 10%/90% ratio. For the evaluation metric, we used the recall@50 averaged over users (Wang et al., 2019; He et al., 2020). For the transductive performance, we report the recall@50 computed over the validation edges. We do not consider the inductive setting for recommender systems. We used the following three datasets in our experiments.

- **gowalla** (Liang et al., 2016; Wang et al., 2019): Given an user-location check-in bipartite graph, the task is to predict new check-in of users.
- **yelp2018** (Wang et al., 2019): Given user-restaurant review graph, the task is to predict new reviews by users.
- **amazon-book** (He & McAuley, 2016; Wang et al., 2019): Given user-product reviews, the task is to predict new reviews by users.

Table 1: Semi-supervised node classification performance with GraphSAGE as the backbone architecture. The metric is classification accuracy. For the "Inductive (cold)", 90% of edges are randomly removed from new nodes. For the results with other edge removal ratios, refer to Table 7 in Appendix. Refer to Table 6 in Appendix for the performance with GCN, where a similar trend is observed.

| Method | arxiv | | | products | | |
|---|---|---|---|---|---|---|
| | Transductive | Inductive | Inductive (cold) | Transductive | Inductive | Inductive (cold) |
| Base | $0.6738_{\pm0.0006}$ | $0.6689_{\pm0.0009}$ | $0.4748_{\pm0.0053}$ | $0.8408_{\pm0.0007}$ | $0.8424_{\pm0.0006}$ | $0.7226_{\pm0.0012}$ |
| DropEdge | $0.6756_{\pm0.0013}$ | $0.6692_{\pm0.0029}$ | $0.5446_{\pm0.0060}$ | $0.8463_{\pm0.0005}$ | $0.8471_{\pm0.0005}$ | $0.7710_{\pm0.0014}$ |
| LocalAug | $0.6824_{\pm0.0004}$ | $\mathbf{0.6773_{\pm0.0023}}$ | $0.4976_{\pm0.0049}$ | $0.8445_{\pm0.0004}$ | $0.8461_{\pm0.0006}$ | $0.7256_{\pm0.0007}$ |
| ColdBrew | $0.6726_{\pm0.0009}$ | $0.6480_{\pm0.0018}$ | $0.5080_{\pm0.0085}$ | $0.8378_{\pm0.0007}$ | $0.8385_{\pm0.0006}$ | $0.7417_{\pm0.0028}$ |
| GraphLessNN | $0.6074_{\pm0.0009}$ | $0.5457_{\pm0.0036}$ | $0.5457_{\pm0.0036}$ | $0.6670_{\pm0.0011}$ | $0.6640_{\pm0.0006}$ | $0.6640_{\pm0.0006}$ |
| Tail-GNN | $0.6613_{\pm0.0016}$ | $0.6557_{\pm0.0022}$ | $0.5389_{\pm0.0023}$ | OOM | OOM | OOM |
| TUNEUP w/o curriculum | $0.6753_{\pm0.0013}$ | $0.6688_{\pm0.0025}$ | $0.5474_{\pm0.0119}$ | $0.8456_{\pm0.0004}$ | $0.8466_{\pm0.0008}$ | $0.7573_{\pm0.0018}$ |
| TUNEUP w/o syn-tails | $0.6785_{\pm0.0011}$ | $\mathbf{0.6758_{\pm0.0008}}$ | $0.4900_{\pm0.0048}$ | $0.8434_{\pm0.0002}$ | $0.8450_{\pm0.0001}$ | $0.7255_{\pm0.0011}$ |
| TUNEUP w/o pseudo-labels | $0.6747_{\pm0.0009}$ | $0.6681_{\pm0.0026}$ | $0.5332_{\pm0.0116}$ | $0.8461_{\pm0.0006}$ | $0.8471_{\pm0.0007}$ | $0.7630_{\pm0.0055}$ |
| **TUNEUP (ours)** | $\mathbf{0.6867_{\pm0.0011}}$ | $\mathbf{0.6784_{\pm0.0032}}$ | $\mathbf{0.6006_{\pm0.0010}}$ | $\mathbf{0.8554_{\pm0.0002}}$ | $\mathbf{0.8564_{\pm0.0004}}$ | $\mathbf{0.8054_{\pm0.0007}}$ |
| Rel. gain over base | +1.9% | +1.4% | +26.5% | +1.7% | +1.7% | +11.5% |

Table 2: Link prediction performance with GraphSAGE as the backbone architecture. The metric is recall@50. For the "Inductive (cold)", 60% of edges are randomly removed from new nodes. For other edge removal ratios, refer to Table 10 in Appendix, where TUNEUP consistently outperforms the baselines. Refer to Table 9 in Appendix for the performance with GCN, where we see a similar trend.

| Method | flickr | | | ppi | | | arxiv | | |
|---|---|---|---|---|---|---|---|---|---|
| | Transductive | Inductive | Inductive (cold) | Transductive | Inductive | Inductive (cold) | Transductive | Inductive | Inductive (cold) |
| Base | $0.1100_{\pm0.0020}$ | $0.1030_{\pm0.0025}$ | $0.0606_{\pm0.0020}$ | $0.1165_{\pm0.0028}$ | $0.1061_{\pm0.0076}$ | $0.0835_{\pm0.0028}$ | $0.1326_{\pm0.0037}$ | $0.1255_{\pm0.0047}$ | $0.0722_{\pm0.0031}$ |
| DropEdge | $0.1437_{\pm0.0008}$ | $0.1297_{\pm0.0011}$ | $0.1007_{\pm0.0017}$ | $0.1750_{\pm0.0017}$ | $0.1492_{\pm0.0073}$ | $0.1282_{\pm0.0042}$ | $0.2015_{\pm0.0036}$ | $0.1753_{\pm0.0040}$ | $0.1204_{\pm0.0027}$ |
| LocalAug | $0.1117_{\pm0.0025}$ | $0.1058_{\pm0.0027}$ | $0.0619_{\pm0.0035}$ | $0.1265_{\pm0.0050}$ | $0.1186_{\pm0.0041}$ | $0.0950_{\pm0.0063}$ | $0.1365_{\pm0.0037}$ | $0.1294_{\pm0.0028}$ | $0.0750_{\pm0.0033}$ |
| ColdBrew | $0.0697_{\pm0.0174}$ | $0.0611_{\pm0.0178}$ | $0.0413_{\pm0.0073}$ | $0.1011_{\pm0.0031}$ | $0.0934_{\pm0.0047}$ | $0.0761_{\pm0.0033}$ | $0.1161_{\pm0.0048}$ | $0.1095_{\pm0.0058}$ | $0.0630_{\pm0.0042}$ |
| Tail-GNN | $0.0950_{\pm0.0014}$ | $0.0804_{\pm0.0014}$ | $0.0746_{\pm0.0018}$ | $0.1150_{\pm0.0011}$ | $0.1002_{\pm0.0025}$ | $0.0929_{\pm0.0030}$ | $0.1081_{\pm0.0025}$ | $0.0866_{\pm0.0039}$ | $0.0643_{\pm0.0039}$ |
| TUNEUP w/o curriculum | $0.1477_{\pm0.0011}$ | $0.1333_{\pm0.0015}$ | $0.1024_{\pm0.0015}$ | $0.1688_{\pm0.0013}$ | $0.1458_{\pm0.0027}$ | $0.1215_{\pm0.0023}$ | $0.1995_{\pm0.0031}$ | $0.1760_{\pm0.0030}$ | $0.1159_{\pm0.0040}$ |
| TUNEUP w/o syn-tails | $0.1101_{\pm0.0021}$ | $0.1029_{\pm0.0024}$ | $0.0603_{\pm0.0017}$ | $0.1172_{\pm0.0031}$ | $0.1057_{\pm0.0071}$ | $0.0824_{\pm0.0021}$ | $0.1341_{\pm0.0028}$ | $0.1273_{\pm0.0031}$ | $0.0738_{\pm0.0021}$ |
| **TUNEUP (ours)** | $\mathbf{0.1508_{\pm0.0023}}$ | $\mathbf{0.1379_{\pm0.0037}}$ | $\mathbf{0.1092_{\pm0.0039}}$ | $\mathbf{0.1819_{\pm0.0017}}$ | $\mathbf{0.1577_{\pm0.0066}}$ | $\mathbf{0.1328_{\pm0.0043}}$ | $\mathbf{0.2101_{\pm0.0041}}$ | $\mathbf{0.1837_{\pm0.0031}}$ | $\mathbf{0.1252_{\pm0.0044}}$ |
| Rel. gain over base | +37.2% | +33.8% | +80.1% | +56.1% | +48.7% | +59.1% | +58.5% | +46.4% | +73.3% |

## 4.2 BASELINES AND ABLATIONS

We compare our TUNEUP against the following strong baselines.

- **Base**: Trains a GNN with the default strategy, *i.e.*, L2–5 of Algorithm 1. The accuracy of pseudo labels coincides with the accuracy of the base GNN.
- **DropEdge** (Wang et al., 2019): Randomly drops edges during training, *i.e.*, L11–15 of Algorithm 1.
- **Local augmentation (LocalAug)** (Liu et al., 2022): Uses a conditional generative model to generate neighboring node features and use them as additional input to a GNN.
- **ColdBrew** (Zheng et al., 2022): Distills head node embeddings computed by the base GNN into an MLP. Uses the resulting MLP to obtain higher-quality tail node embeddings.
- **GraphLessNN** (Zhang et al., 2022): Distills the pseudo labels predicted by the base GNN into an MLP. Uses the resulting MLP to make prediction.
- **Tail-GNN** (Liu et al., 2021): Adds a tail-node specific component inside the original GNN.
- **RAWLS-GCN** (Kang et al., 2022): Modifies the GCN's adjacency matrix to be doubly-stochastic (*i.e.*, all rows and columns sum to 1).

Note that GraphLessNN is only applicable for node classification. LocalAug and ColdBrew require input node features to be available; hence, not applicable to recommender systems. RAWLS-GCN is only applicable to the GCN architecture.

In addition to the existing baselines, we consider the following three direct ablations of TUNEUP.

- **TUNEUP w/o curriculum**: Interleaves the first stage prediction (L3 in Algorithm 1) and the second stage prediction (L12–13 in Algorithm 1) in every parameter update. It is close to TUNEUP except that it does not follow the two-stage curriculum learning strategy.
- **TUNEUP w/o syn-tails**: No L12 in Algorithm 1.
- **TUNEUP w/o pseudo-labels**: No L8 in Algorithm 1.

Table 3: Transductive performance on the recommender systems datasets. The metric is recall@50.

| Method | gowalla | | yelp2018 | | amazon-book | |
|---|---|---|---|---|---|---|
| | SAGE | GCN | SAGE | GCN | SAGE | GCN |
| Base | $0.0856_{\pm0.0005}$ | $0.0911_{\pm0.0006}$ | $\mathbf{0.0864}_{\pm\mathbf{0.0005}}$ | $0.0791_{\pm0.0003}$ | $0.0559_{\pm0.0003}$ | $0.0540_{\pm0.0002}$ |
| DropEdge | $0.0835_{\pm0.0004}$ | $0.0827_{\pm0.0012}$ | $0.0814_{\pm0.0003}$ | $0.0719_{\pm0.0013}$ | $0.0543_{\pm0.0003}$ | $0.0556_{\pm0.0004}$ |
| Tail-GNN | $0.0806_{\pm0.0005}$ | $0.0815_{\pm0.0016}$ | $0.0713_{\pm0.0004}$ | $0.0713_{\pm0.0005}$ | $0.0562_{\pm0.0002}$ | $0.0542_{\pm0.0003}$ |
| RAWS-GCN | – | $0.0632_{\pm0.0006}$ | – | $0.0684_{\pm0.0004}$ | – | $0.0476_{\pm0.0002}$ |
| TuneUp w/o curriculum | $0.0849_{\pm0.0004}$ | $0.0874_{\pm0.0038}$ | $0.0837_{\pm0.0006}$ | $0.0755_{\pm0.0021}$ | $0.0549_{\pm0.0001}$ | $0.0539_{\pm0.0004}$ |
| TuneUp w/o syn-tails | $0.0859_{\pm0.0006}$ | $0.0917_{\pm0.0005}$ | $\mathbf{0.0866}_{\pm\mathbf{0.0005}}$ | $0.0797_{\pm0.0003}$ | $0.0561_{\pm0.0003}$ | $0.0542_{\pm0.0002}$ |
| **TuneUp (ours)** | $\mathbf{0.1043}_{\pm\mathbf{0.0020}}$ | $\mathbf{0.1102}_{\pm\mathbf{0.0013}}$ | $\mathbf{0.0866}_{\pm\mathbf{0.0004}}$ | $\mathbf{0.0885}_{\pm\mathbf{0.0006}}$ | $\mathbf{0.0577}_{\pm\mathbf{0.0005}}$ | $\mathbf{0.0630}_{\pm\mathbf{0.0004}}$ |
| Relative gain over base | +21.8% | +21.0% | +0.2% | +11.9% | +3.1% | +16.7% |

Table 4: The improvement with TuneUp over the base GNNs for diverse GNN model architectures. We used the same set of datasets as Figure 1. [†]For semi-supervised node classification, GAT gave Out-Of-Memory (OOM) on the products dataset; so we report the performance on arxiv instead.

| Task | Config | Setting | SAGE(-mean) | GCN | SAGE-max | SAGE-sum | GAT |
|---|---|---|---|---|---|---|---|
| Semi-sup node classification (products) | Transductive | Base | $0.8408_{\pm0.0007}$ | $0.8432_{\pm0.0007}$ | $0.8134_{\pm0.0008}$ | $\mathbf{0.7607}_{\pm\mathbf{0.0029}}$ | OOM / $0.6864_{\pm0.0021}$[†] |
| | | TuneUp | $\mathbf{0.8554}_{\pm\mathbf{0.0002}}$ | $\mathbf{0.8523}_{\pm\mathbf{0.0007}}$ | $\mathbf{0.8371}_{\pm\mathbf{0.0008}}$ | $\mathbf{0.7615}_{\pm\mathbf{0.0036}}$ | OOM / $\mathbf{0.6968}_{\pm\mathbf{0.0010}}$[†] |
| | Inductive | Base | $0.8424_{\pm0.0006}$ | $0.8447_{\pm0.0008}$ | $0.8133_{\pm0.0011}$ | $\mathbf{0.7613}_{\pm\mathbf{0.0029}}$ | OOM / $0.6800_{\pm0.0026}$[†] |
| | | TuneUp | $\mathbf{0.8564}_{\pm\mathbf{0.0004}}$ | $\mathbf{0.8535}_{\pm\mathbf{0.0005}}$ | $\mathbf{0.8369}_{\pm\mathbf{0.0012}}$ | $\mathbf{0.7619}_{\pm\mathbf{0.0048}}$ | OOM / $\mathbf{0.6936}_{\pm\mathbf{0.0009}}$[†] |
| | Inductive (cold) | Base | $0.7226_{\pm0.0012}$ | $0.7461_{\pm0.0034}$ | $0.6908_{\pm0.0007}$ | $0.5315_{\pm0.0059}$ | OOM / $0.5411_{\pm0.0035}$[†] |
| | | TuneUp | $\mathbf{0.8054}_{\pm\mathbf{0.0007}}$ | $\mathbf{0.7925}_{\pm\mathbf{0.0049}}$ | $\mathbf{0.7873}_{\pm\mathbf{0.0016}}$ | $\mathbf{0.5490}_{\pm\mathbf{0.0129}}$ | OOM / $\mathbf{0.5932}_{\pm\mathbf{0.0053}}$[†] |
| Link prediction (arxiv) | Transductive | Base | $0.1326_{\pm0.0037}$ | $0.2152_{\pm0.0014}$ | $0.1618_{\pm0.0033}$ | $0.0730_{\pm0.0017}$ | $0.2295_{\pm0.0035}$ |
| | | TuneUp | $\mathbf{0.2101}_{\pm\mathbf{0.0041}}$ | $\mathbf{0.2417}_{\pm\mathbf{0.0018}}$ | $\mathbf{0.2397}_{\pm\mathbf{0.0031}}$ | $\mathbf{0.1168}_{\pm\mathbf{0.0056}}$ | $\mathbf{0.2569}_{\pm\mathbf{0.0014}}$ |
| | Inductive | Base | $0.1255_{\pm0.0047}$ | $0.2077_{\pm0.0014}$ | $0.1503_{\pm0.0026}$ | $0.0703_{\pm0.0023}$ | $0.2080_{\pm0.0029}$ |
| | | TuneUp | $\mathbf{0.1837}_{\pm\mathbf{0.0031}}$ | $\mathbf{0.2254}_{\pm\mathbf{0.0022}}$ | $\mathbf{0.2140}_{\pm\mathbf{0.0040}}$ | $\mathbf{0.1031}_{\pm\mathbf{0.0053}}$ | $\mathbf{0.2360}_{\pm\mathbf{0.0022}}$ |
| | Inductive (cold) | Base | $0.0722_{\pm0.0031}$ | $0.1192_{\pm0.0019}$ | $0.0998_{\pm0.0031}$ | $0.0547_{\pm0.0018}$ | $0.1316_{\pm0.0034}$ |
| | | TuneUp | $\mathbf{0.1252}_{\pm\mathbf{0.0044}}$ | $\mathbf{0.1440}_{\pm\mathbf{0.0017}}$ | $\mathbf{0.1559}_{\pm\mathbf{0.0030}}$ | $\mathbf{0.0788}_{\pm\mathbf{0.0056}}$ | $\mathbf{0.1627}_{\pm\mathbf{0.0034}}$ |
| Recsys (gowalla) | Transductive | Base | $0.0856_{\pm0.0005}$ | $0.0911_{\pm0.0006}$ | $0.0870_{\pm0.0005}$ | $0.0896_{\pm0.0003}$ | $0.0814_{\pm0.0005}$ |
| | | TuneUp | $\mathbf{0.1043}_{\pm\mathbf{0.0020}}$ | $\mathbf{0.1102}_{\pm\mathbf{0.0013}}$ | $\mathbf{0.1063}_{\pm\mathbf{0.0026}}$ | $\mathbf{0.1129}_{\pm\mathbf{0.0025}}$ | $\mathbf{0.0834}_{\pm\mathbf{0.0008}}$ |

Another possible ablation, TuneUp w/o the first stage training (*i.e.*, only performing the second stage training of L2–5 in Algorithm 1), is essentially covered as DropEdge in our experiments.

## 4.3 GNN Model Architectures

We mainly experimented with two classical yet strong GNN models: the mean-pooling variant of GraphSAGE (or SAGE for short) (Hamilton et al., 2017) and GCN (Kipf & Welling, 2017). In Table 4, we additionally experimented with the max- and sum-pooling variants of GraphSAGE as well as the Graph Attention Network (GAT) (Velickovic et al., 2018) to demonstrate the applicability of TuneUp on diverse GNN architectures to improve their performance. In total, we have five diverse GNN architectures that cover representative aggregation schemes (*i.e.*, mean, renormalized-mean (Kipf & Welling, 2017), max, sum, and attention) that many recent advanced GNN architectures are based on (Corso et al., 2020; Shi et al., 2021; You et al., 2020b; Wu et al., 2019; Rossi et al., 2020; Li et al., 2018; You et al., 2020a).

## 4.4 Hyper-parameters

We used 3-layer GNNs and the Adam optimizer (Kingma & Ba, 2015) for all GNN models and datasets, which we found to perform well in our preliminary experiments. For all methods, we performed the early stopping and tuned their hyper-parameters based on the transductive validation performance. We used the resulting models for both transductive and inductive prediction. For the drop edge ratio $\alpha$, we tuned it from [0.25, 0.5, 0.75] for all the datasets. We repeated all the experiments with 5 different training seeds to report the mean and the standard deviation. More details are described in Appendix A.

## 4.5 Results

We first compare TuneUp against the base GNNs that are trained with the default strategy. The last rows of Tables 1, 2, and 3 highlight the relative improvement of TuneUp over the base GNNs. TuneUp improves over the base GNNs across the transductive settings, giving up to 1.9%, 58.5%,

and 21.8% relative improvement in the semi-supervised node classification, link prediction, and recommender systems, respectively. TUNEUP gives even larger improvement on the challenging cold-start inductive prediction setting, yielding up to 26.5% and 80.1% relative improvement on the node classification and link prediction, respectively. In Appendix, we provide Tables 7, 8, 10, and 11 to show the full results on the cold-start inductive prediction with the different edge removal ratios from new nodes. The larger the ratio is, the more cold-start the setting becomes. We observe that TUNEUP provides larger relative gain on larger edge removal ratios, demonstrating its effectiveness on the highly cold-start prediction setting.

We also analyze the degree-specific generalization improvement and highlight the results in Figure 1. The full results (two GNN architectures times the eight datasets) are available in Figures 2, 3, and 4 in Appendix. Across all datasets, architectures, and node degrees, TUNEUP produces consistent improvement over the base GNNs. Not surprisingly, improvement is most significant on tail nodes.

Finally, we compare TUNEUP against the strong baselines and ablation methods described in Section 4.2. Tables 1, 2, and 3 show the results. We summarize our findings below.

- TUNEUP outperforms the graph augmentation methods (DropEdge and LocalAug) as well as the specialized methods for tail nodes (ColdBrew, GraphLessNN, and Tail-GNN), establishing its superior performance against the existing strong baseline methods.
- TUNEUP outperforms TUNEUP w/o curriculum, which highlights the importance of the two-stage curriculum learning strategy in TUNEUP.
- TUNEUP also outperforms TUNEUP w/o syn-tails and TUNEUP w/o pseudo-labels, which suggests that both of the ablated components are necessary for TUNEUP to achieve the high performance.
- On semi-supervised node classification (Table 1), TUNEUP w/o syn-tails, *i.e.*, conventional semi-supervised training with the pseudo labels (Lee et al., 2013), gives limited improvement over the base GNN. In contrast, TUNEUP trains the GNN to predict pseudo labels *with limited neighborhood*, which gives significant improvement over the base GNN. Moreover, TUNEUP significantly outperforms DropEdge, suggesting the importance of using DropEdge together with pseudo labels.
- On link prediction (Table 2), DropEdge (TUNEUP without the first stage) already gives significant performance improvement over the base GNN, implying the unrealized potential of DropEdge on this task, beyond node classification (Rong et al., 2020). Nonetheless, TUNEUP still gives consistent improvement over DropEdge, suggesting the benefit of the two-stage training.
- On recommender systems (Table 3), TUNEUP is the only method that produced the significantly better performance than the base GNN. DropEdge and TUNEUP w/ curriculum even gave worse performance than the base GNN. This is possibly because jointly learning the GNN and shallow embeddings is hard without the two-stage training.
- Overall, TUNEUP, despite its simplicity, is the only method that yielded consistent improvement *across* the three prediction tasks.
- From Table 4, we see that TUNEUP improves the performance on the five *diverse GNN architectures*. Although the performance improvement with the sum aggregation is limited for semi-supervised node classification, the sum aggregation gave the poor base GNN performance anyway due to the poor inductive bias and unstable training (Wu et al., 2019; Hamilton et al., 2017).

## 5  CONCLUSIONS

In this paper, we presented TUNEUP, a curriculum learning strategy to train a GNN to improve its generalization performance. TUNEUP first trains a GNN to produce a strong base GNN that performs well on easy head nodes. It then proceeds to improve the prediction over hard tail nodes by finetuning the base GNN with additional synthetic tail nodes. TUNEUP is a general strategy that can be used to train any GNN model with any supervised loss. Through extensive experiments, we demonstrated the effectiveness of TUNEUP on a wide range of settings, including five GNN architectures, three types of prediction tasks, and both transductive and inductive settings. Overall, our work suggests that training strategies matter in improving generalization of GNNs and can be complementary to advances in model architectures and task-specific losses.

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

Table 5: Statistics of nodes used for the transductive evaluation. For link prediction and the recommender system graphs (user-item bipartite graphs), we only evaluate on nodes/users that have at least one edge in the validation set.

| Task | Dataset | #Nodes | Avg deg. | Feat. dim |
|------|---------|--------|----------|-----------|
| Node cls. | arxiv | 143,941 | 12.93 | 128 |
| | products | 2,277,597 | 48.01 | 100 |
| Link pred. | flickr | 82,981 | 4.81 | 500 |
| | ppi | 15,390 | 17.30 | 1,280 |
| | arxiv | 141,917 | 7.20 | 128 |
| Recsys | gowalla | 29,858 | 3.44 | – |
| | yelp2018 | 31,668 | 4.93 | – |
| | amazon-book | 52,643 | 5.67 | – |

Table 6: Semi-supervised node classification performance with GCN as the backbone architecture. The evaluation metric is classification accuracy. For the "Inductive (cold)", 90% of edges are randomly removed from new nodes. For the results with other edge removal ratios, refer to Table 8 in Appendix.

| Method | arxiv | | | products | | |
|--------|-------|------|----------|----------|------|----------|
| | Transductive | Inductive | Inductive (cold) | Transductive | Inductive | Inductive (cold) |
| Base | $0.6922_{\pm 0.0003}$ | $0.6893_{\pm 0.0021}$ | $0.5497_{\pm 0.0038}$ | $0.8432_{\pm 0.0007}$ | $0.8447_{\pm 0.0008}$ | $0.7461_{\pm 0.0034}$ |
| DropEdge | $0.6961_{\pm 0.0008}$ | $0.6939_{\pm 0.0011}$ | $0.5630_{\pm 0.0019}$ | $0.8486_{\pm 0.0003}$ | $0.8497_{\pm 0.0007}$ | $0.7671_{\pm 0.0020}$ |
| LocalAug | $\mathbf{0.6987}_{\pm 0.0012}$ | $\mathbf{0.6962}_{\pm 0.0023}$ | $0.5660_{\pm 0.0017}$ | $0.8488_{\pm 0.0004}$ | $0.8501_{\pm 0.0005}$ | $0.7526_{\pm 0.0011}$ |
| ColdBrew | $0.6867_{\pm 0.0009}$ | $0.6732_{\pm 0.0023}$ | $0.5369_{\pm 0.0039}$ | $0.8399_{\pm 0.0008}$ | $0.8406_{\pm 0.0007}$ | $0.7371_{\pm 0.0025}$ |
| GraphLessNN | $0.6129_{\pm 0.0012}$ | $0.5462_{\pm 0.0028}$ | $0.5462_{\pm 0.0028}$ | $0.6669_{\pm 0.0007}$ | $0.6648_{\pm 0.0007}$ | $0.6648_{\pm 0.0007}$ |
| RAWLS-GCN | $0.6708_{\pm 0.0013}$ | $0.6698_{\pm 0.0025}$ | $0.5324_{\pm 0.0025}$ | $0.8209_{\pm 0.0010}$ | $0.8220_{\pm 0.0011}$ | $0.7096_{\pm 0.0053}$ |
| Tail-GNN | $0.6437_{\pm 0.0008}$ | $0.6423_{\pm 0.0020}$ | $0.5371_{\pm 0.0043}$ | OOM | OOM | OOM |
| TUNEUP w/o curriculum | $0.6961_{\pm 0.0008}$ | $0.6924_{\pm 0.0027}$ | $0.5593_{\pm 0.0032}$ | $0.8481_{\pm 0.0005}$ | $0.8490_{\pm 0.0003}$ | $0.7624_{\pm 0.0030}$ |
| TUNEUP w/o syn-tails | $0.6940_{\pm 0.0005}$ | $0.6924_{\pm 0.0020}$ | $0.5607_{\pm 0.0037}$ | $0.8452_{\pm 0.0006}$ | $0.8468_{\pm 0.0006}$ | $0.7549_{\pm 0.0026}$ |
| TUNEUP w/o pseudo-labels | $0.6967_{\pm 0.0009}$ | $0.6936_{\pm 0.0015}$ | $0.5534_{\pm 0.0039}$ | $0.8488_{\pm 0.0007}$ | $0.8497_{\pm 0.0003}$ | $0.7676_{\pm 0.0024}$ |
| **TUNEUP (ours)** | $\mathbf{0.6988}_{\pm 0.0006}$ | $\mathbf{0.6973}_{\pm 0.0014}$ | $\mathbf{0.5923}_{\pm 0.0053}$ | $\mathbf{0.8523}_{\pm 0.0007}$ | $\mathbf{0.8535}_{\pm 0.0005}$ | $\mathbf{0.7925}_{\pm 0.0049}$ |
| Relative gain over base | +1.0% | +1.1% | +7.8% | +1.1% | +1.0% | +6.2% |

## A    DETAILS OF HYPER-PARAMETERS

Here we present the details of hyper-parameters we used in our experiments.

**Semi-supervised node classification**. We used the hidden dimensionality of 256 and 64 for arxiv and products, respectively. We trained GNNs in a full-batch manner, and for products, we used the reduced dimensionality of 64 so that the entire graph can be fit into the limited GPU memory of 45GB. Mini-batch training is left for future work. We used 1500 epochs for both the default training and finetuning. The learning rate is set to 0.001.

**Link prediction**. We used the hidden dimensionality of 256 for all datasets. We added the L2 regularization on the node embeddings and tuned its weight for each dataset and GNN architecture. For both the default training and finetuning, we used 1000 epochs and the learning rate of 0.0001.

**Recommender systems**. We used the shallow embedding dimensionality of 64 and the hidden embedding dimensionality of 256. Similar to the link prediction, we added the L2 regularization to the node embeddings and tuned its weight for each dataset and GNN architecture. For the default training, we trained the model for 2000 epochs with the initial learning rate of 0.001, which is multiplied by 0.1 at the 1000th and 1500th epoch. For finetuning, we used 500 epochs with the learning rate of 0.0001.

For training strategies without curriculum learning, we used the same configuration as the default training.

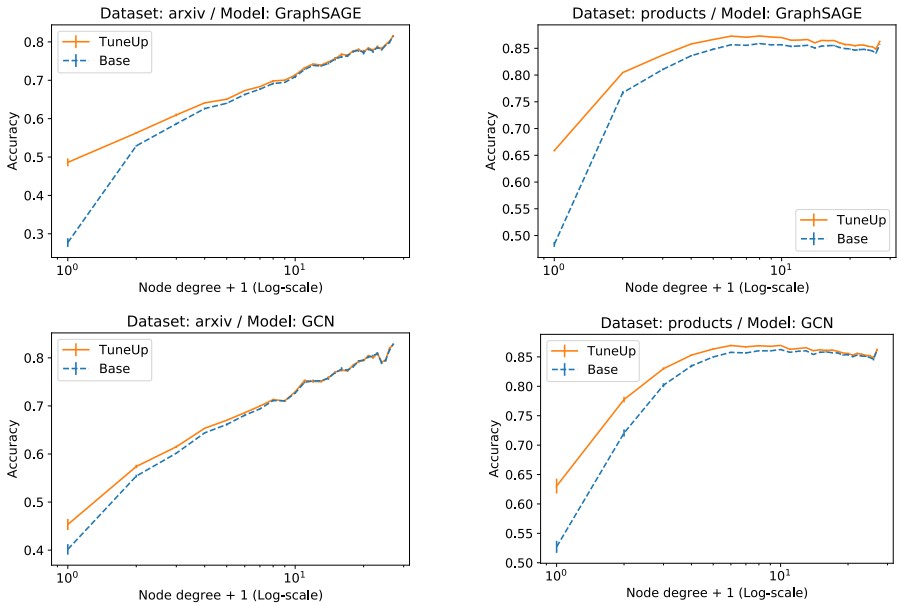

Figure 2: Degree-specific generalization performance of the base GNN and TUNEUP in transductive semi-supervised node classification. The evaluation metric is classification accuracy.

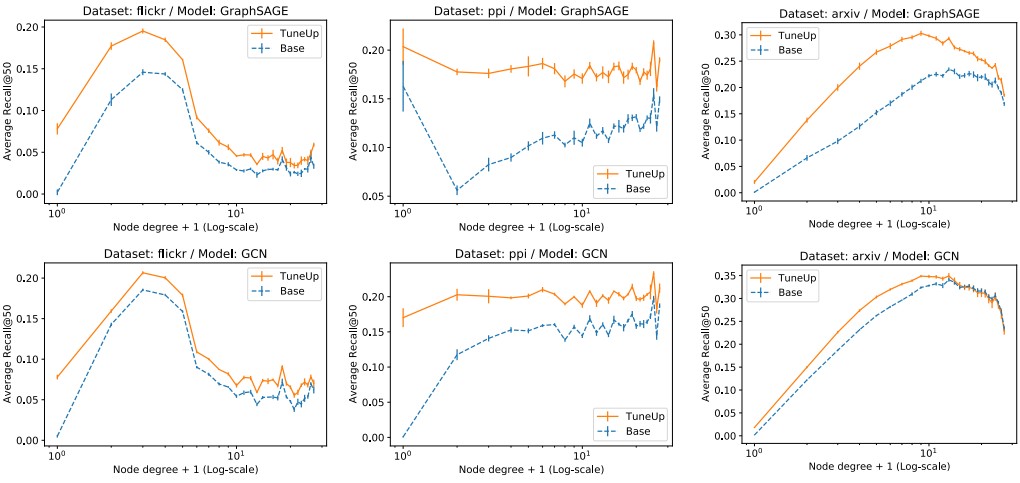

Figure 3: Degree-specific generalization performance of the base GNN and TUNEUP in transductive link prediction. The evaluation metric is recall@50.

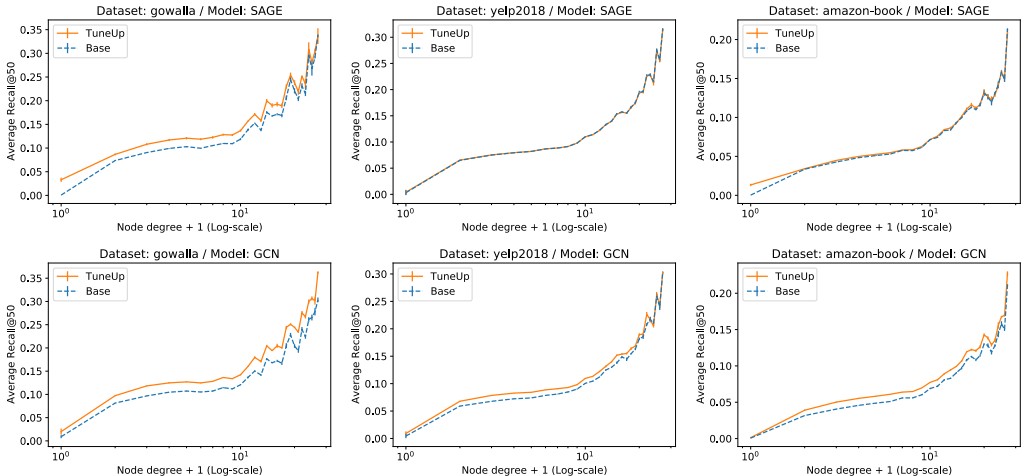

Figure 4: Degree-specific generalization performance of the base GNN and TUNEUP in transductive recommender systems. The evaluation metric is recall@50.

Table 7: Cold-start inductive node classification performance with GraphSAGE as the backbone architecture. The larger the edge removal ratio is, the more cold-start the prediction task becomes. The evaluation metric is classification accuracy.

| Method | arxiv | | | products | | |
|---|---|---|---|---|---|---|
| | Edge removal ratio | | | Edge removal ratio | | |
| | 30% | 60% | 90% | 30% | 60% | 90% |
| Base | $0.6452_{\pm 0.0017}$ | $0.6000_{\pm 0.0012}$ | $0.4748_{\pm 0.0053}$ | $0.8334_{\pm 0.0010}$ | $0.8129_{\pm 0.0009}$ | $0.7226_{\pm 0.0012}$ |
| DropEdge | $0.6532_{\pm 0.0019}$ | $0.6247_{\pm 0.0040}$ | $0.5446_{\pm 0.0060}$ | $0.8409_{\pm 0.0005}$ | $0.8280_{\pm 0.0005}$ | $0.7710_{\pm 0.0014}$ |
| LocalAug | $0.6552_{\pm 0.0024}$ | $0.6158_{\pm 0.0014}$ | $0.4976_{\pm 0.0049}$ | $0.8370_{\pm 0.0008}$ | $0.8166_{\pm 0.0008}$ | $0.7256_{\pm 0.0007}$ |
| ColdBrew | $0.6289_{\pm 0.0022}$ | $0.5943_{\pm 0.0042}$ | $0.5080_{\pm 0.0085}$ | $0.8314_{\pm 0.0009}$ | $0.8142_{\pm 0.0010}$ | $0.7417_{\pm 0.0028}$ |
| GraphLessNN | $0.5457_{\pm 0.0036}$ | $0.5457_{\pm 0.0036}$ | $0.5457_{\pm 0.0036}$ | $0.6640_{\pm 0.0006}$ | $0.6640_{\pm 0.0006}$ | $0.6640_{\pm 0.0006}$ |
| Tail-GNN | $0.6393_{\pm 0.0023}$ | $0.6131_{\pm 0.0017}$ | $0.5389_{\pm 0.0023}$ | OOM | OOM | OOM |
| TUNEUP w/o curriculum | $0.6530_{\pm 0.0026}$ | $0.6270_{\pm 0.0036}$ | $0.5474_{\pm 0.0119}$ | $0.8395_{\pm 0.0006}$ | $0.8243_{\pm 0.0006}$ | $0.7573_{\pm 0.0018}$ |
| TUNEUP w/o syn-tails | $0.6523_{\pm 0.0020}$ | $0.6103_{\pm 0.0024}$ | $0.4900_{\pm 0.0048}$ | $0.8361_{\pm 0.0004}$ | $0.8161_{\pm 0.0005}$ | $0.7255_{\pm 0.0011}$ |
| TUNEUP w/o pseudo-labels | $0.6505_{\pm 0.0015}$ | $0.6198_{\pm 0.0051}$ | $0.5332_{\pm 0.0116}$ | $0.8404_{\pm 0.0009}$ | $0.8260_{\pm 0.0015}$ | $0.7630_{\pm 0.0055}$ |
| **TUNEUP (ours)** | $\mathbf{0.6673}_{\pm 0.0018}$ | $\mathbf{0.6507}_{\pm 0.0027}$ | $\mathbf{0.6006}_{\pm 0.0010}$ | $\mathbf{0.8523}_{\pm 0.0004}$ | $\mathbf{0.8434}_{\pm 0.0003}$ | $\mathbf{0.8054}_{\pm 0.0007}$ |
| Relative gain over base | +3.4% | +8.4% | +26.5% | +2.3% | +3.8% | +11.5% |

Table 8: Cold-start inductive node classification performance with GCN as the backbone architecture. The larger the edge removal ratio is, the more cold-start the prediction task becomes. The evaluation metric is classification accuracy.

| Method | arxiv | | | products | | |
|---|---|---|---|---|---|---|
| | Edge removal ratio | | | Edge removal ratio | | |
| | 30% | 60% | 90% | 30% | 60% | 90% |
| Base | $0.6715_{\pm 0.0018}$ | $0.6400_{\pm 0.0028}$ | $0.5497_{\pm 0.0038}$ | $0.8375_{\pm 0.0008}$ | $0.8210_{\pm 0.0011}$ | $0.7461_{\pm 0.0034}$ |
| DropEdge | $0.6753_{\pm 0.0021}$ | $0.6475_{\pm 0.0031}$ | $0.5630_{\pm 0.0019}$ | $0.8438_{\pm 0.0006}$ | $0.8301_{\pm 0.0009}$ | $0.7671_{\pm 0.0020}$ |
| LocalAug | $0.6785_{\pm 0.0018}$ | $0.6491_{\pm 0.0021}$ | $0.5660_{\pm 0.0017}$ | $0.8429_{\pm 0.0003}$ | $0.8267_{\pm 0.0004}$ | $0.7526_{\pm 0.0011}$ |
| ColdBrew | $0.6505_{\pm 0.0020}$ | $0.6180_{\pm 0.0037}$ | $0.5369_{\pm 0.0039}$ | $0.8330_{\pm 0.0009}$ | $0.8156_{\pm 0.0011}$ | $0.7371_{\pm 0.0025}$ |
| GraphLessNN | $0.5462_{\pm 0.0028}$ | $0.5462_{\pm 0.0028}$ | $0.5462_{\pm 0.0028}$ | $0.6648_{\pm 0.0007}$ | $0.6648_{\pm 0.0007}$ | $0.6648_{\pm 0.0007}$ |
| RAWLS-GCN | $0.6488_{\pm 0.0015}$ | $0.6115_{\pm 0.0026}$ | $0.5324_{\pm 0.0025}$ | $0.8124_{\pm 0.0015}$ | $0.7916_{\pm 0.0022}$ | $0.7096_{\pm 0.0053}$ |
| Tail-GNN | $0.6283_{\pm 0.0018}$ | $0.6032_{\pm 0.0013}$ | $0.5371_{\pm 0.0043}$ | OOM | OOM | OOM |
| TUNEUP w/o curriculum | $0.6755_{\pm 0.0012}$ | $0.6448_{\pm 0.0023}$ | $0.5593_{\pm 0.0032}$ | $0.8427_{\pm 0.0005}$ | $0.8285_{\pm 0.0007}$ | $0.7624_{\pm 0.0030}$ |
| TUNEUP w/o syn-tails | $0.6741_{\pm 0.0018}$ | $0.6468_{\pm 0.0025}$ | $0.5607_{\pm 0.0037}$ | $0.8398_{\pm 0.0006}$ | $0.8246_{\pm 0.0011}$ | $0.7549_{\pm 0.0026}$ |
| TUNEUP w/o pseudo-labels | $0.6750_{\pm 0.0011}$ | $0.6439_{\pm 0.0014}$ | $0.5534_{\pm 0.0039}$ | $0.8439_{\pm 0.0009}$ | $0.8300_{\pm 0.0010}$ | $0.7676_{\pm 0.0024}$ |
| **TUNEUP (ours)** | $\mathbf{0.6808}_{\pm 0.0019}$ | $\mathbf{0.6604}_{\pm 0.0007}$ | $\mathbf{0.5923}_{\pm 0.0053}$ | $\mathbf{0.8489}_{\pm 0.0006}$ | $\mathbf{0.8385}_{\pm 0.0013}$ | $\mathbf{0.7925}_{\pm 0.0049}$ |
| Relative gain over base | +1.4% | +3.2% | +7.8% | +1.4% | +2.1% | +6.2% |

Table 9: Link prediction performance with the GCN as the backbone architecture. The evaluation metric is recall@50. For the "Inductive (cold)", 60% of edges are randomly removed from new nodes. For the results with other edge removal ratios, refer to Table 11 in Appendix.

| Method | flickr | | | ppi | | | arxiv | | |
|---|---|---|---|---|---|---|---|---|---|
| | Transductive | Inductive | Inductive (cold) | Transductive | Inductive | Inductive (cold) | Transductive | Inductive | Inductive (cold) |
| Base | $0.1431_{\pm 0.0011}$ | $0.1369_{\pm 0.0022}$ | $0.0882_{\pm 0.0022}$ | $0.1491_{\pm 0.0020}$ | $0.1178_{\pm 0.0033}$ | $0.0909_{\pm 0.0037}$ | $0.2152_{\pm 0.0014}$ | $0.2077_{\pm 0.0014}$ | $0.1192_{\pm 0.0019}$ |
| DropEdge | $0.1543_{\pm 0.0007}$ | $0.1425_{\pm 0.0033}$ | $0.1006_{\pm 0.0020}$ | $0.1903_{\pm 0.0009}$ | $0.1712_{\pm 0.0096}$ | $\mathbf{0.1364}_{\pm 0.0066}$ | $0.2306_{\pm 0.0012}$ | $0.2116_{\pm 0.0017}$ | $0.1346_{\pm 0.0010}$ |
| LocalAug | $0.1487_{\pm 0.0017}$ | $0.1445_{\pm 0.0020}$ | $0.0936_{\pm 0.0030}$ | $0.1460_{\pm 0.0030}$ | $0.1321_{\pm 0.0062}$ | $0.0920_{\pm 0.0046}$ | $0.2210_{\pm 0.0010}$ | $0.2134_{\pm 0.0008}$ | $0.1216_{\pm 0.0015}$ |
| ColdBrew | $0.1201_{\pm 0.0019}$ | $0.1154_{\pm 0.0019}$ | $0.0686_{\pm 0.0035}$ | $0.1133_{\pm 0.0082}$ | $0.0383_{\pm 0.0272}$ | $0.0376_{\pm 0.0223}$ | $0.1783_{\pm 0.0017}$ | $0.1723_{\pm 0.0017}$ | $0.0927_{\pm 0.0015}$ |
| Tail-GNN | $0.1282_{\pm 0.0007}$ | $0.1230_{\pm 0.0019}$ | $0.0830_{\pm 0.0030}$ | $0.1564_{\pm 0.0016}$ | $0.1294_{\pm 0.0088}$ | $0.0971_{\pm 0.0053}$ | $0.1253_{\pm 0.0005}$ | $0.1144_{\pm 0.0014}$ | $0.0811_{\pm 0.0014}$ |
| RAWLS-GCN | $0.0704_{\pm 0.0015}$ | $0.0448_{\pm 0.0018}$ | $0.0430_{\pm 0.0017}$ | $0.1280_{\pm 0.0017}$ | $0.0653_{\pm 0.0045}$ | $0.0453_{\pm 0.0010}$ | $0.1010_{\pm 0.0009}$ | $0.0816_{\pm 0.0010}$ | $0.0408_{\pm 0.0014}$ |
| TUNEUP w/o curriculum | $0.1571_{\pm 0.0007}$ | $0.1466_{\pm 0.0011}$ | $0.1048_{\pm 0.0016}$ | $0.1827_{\pm 0.0020}$ | $0.1511_{\pm 0.0048}$ | $0.1241_{\pm 0.0039}$ | $0.2341_{\pm 0.0007}$ | $0.2169_{\pm 0.0008}$ | $0.1356_{\pm 0.0017}$ |
| TUNEUP w/o syn-tails | $0.1458_{\pm 0.0010}$ | $0.1398_{\pm 0.0016}$ | $0.0883_{\pm 0.0015}$ | $0.1553_{\pm 0.0012}$ | $0.1240_{\pm 0.0091}$ | $0.0916_{\pm 0.0053}$ | $0.2197_{\pm 0.0009}$ | $0.2123_{\pm 0.0010}$ | $0.1195_{\pm 0.0014}$ |
| **TUNEUP (ours)** | $\mathbf{0.1637}_{\pm 0.0015}$ | $\mathbf{0.1517}_{\pm 0.0008}$ | $\mathbf{0.1080}_{\pm 0.0027}$ | $\mathbf{0.2009}_{\pm 0.0012}$ | $\mathbf{0.1857}_{\pm 0.0036}$ | $\mathbf{0.1430}_{\pm 0.0073}$ | $\mathbf{0.2417}_{\pm 0.0018}$ | $\mathbf{0.2254}_{\pm 0.0022}$ | $\mathbf{0.1440}_{\pm 0.0017}$ |
| Relative gain over base | +14.4% | +10.8% | +22.5% | +34.8% | +57.7% | +57.4% | +12.3% | +8.5% | +20.9% |

Table 10: Cold-start inductive link prediction performance with GraphSAGE. The evaluation metric is recall@50. The larger the edge removal ratio is, the more cold-start the prediction task becomes.

| Method | flickr | | | ppi | | | arxiv | | |
|---|---|---|---|---|---|---|---|---|---|
| | Edge removal ratio | | | Edge removal ratio | | | Edge removal ratio | | |
| | 30% | 60% | 90% | 30% | 60% | 90% | 30% | 60% | 90% |
| Base | $0.0830_{\pm 0.0021}$ | $0.0606_{\pm 0.0020}$ | $0.0186_{\pm 0.0026}$ | $0.0990_{\pm 0.0058}$ | $0.0835_{\pm 0.0028}$ | $0.0708_{\pm 0.0035}$ | $0.1021_{\pm 0.0046}$ | $0.0722_{\pm 0.0031}$ | $0.0240_{\pm 0.0025}$ |
| DropEdge | $0.1159_{\pm 0.0015}$ | $0.1007_{\pm 0.0017}$ | $0.0602_{\pm 0.0037}$ | $0.1401_{\pm 0.0072}$ | $0.1282_{\pm 0.0042}$ | $\mathbf{0.0999}_{\pm 0.0043}$ | $0.1531_{\pm 0.0045}$ | $0.1204_{\pm 0.0027}$ | $\mathbf{0.0579}_{\pm 0.0019}$ |
| LocalAug | $0.0855_{\pm 0.0028}$ | $0.0619_{\pm 0.0035}$ | $0.0185_{\pm 0.0026}$ | $0.1110_{\pm 0.0049}$ | $0.0950_{\pm 0.0063}$ | $0.0717_{\pm 0.0162}$ | $0.1051_{\pm 0.0031}$ | $0.0750_{\pm 0.0033}$ | $0.0261_{\pm 0.0045}$ |
| ColdBrew | $0.0497_{\pm 0.0129}$ | $0.0413_{\pm 0.0073}$ | $0.0356_{\pm 0.0086}$ | $0.0846_{\pm 0.0038}$ | $0.0761_{\pm 0.0033}$ | $0.0597_{\pm 0.0077}$ | $0.0887_{\pm 0.0042}$ | $0.0630_{\pm 0.0042}$ | $0.0318_{\pm 0.0045}$ |
| Tail-GNN | $0.0767_{\pm 0.0011}$ | $0.0746_{\pm 0.0018}$ | $0.0521_{\pm 0.0018}$ | $0.0939_{\pm 0.0032}$ | $0.0929_{\pm 0.0030}$ | $0.0842_{\pm 0.0072}$ | $0.0771_{\pm 0.0037}$ | $0.0643_{\pm 0.0039}$ | $0.0410_{\pm 0.0034}$ |
| TUNEUP w/o curriculum | $0.1192_{\pm 0.0015}$ | $0.1024_{\pm 0.0015}$ | $0.0624_{\pm 0.0023}$ | $0.1374_{\pm 0.0017}$ | $0.1215_{\pm 0.0023}$ | $0.0905_{\pm 0.0053}$ | $0.1516_{\pm 0.0038}$ | $0.1159_{\pm 0.0040}$ | $\mathbf{0.0567}_{\pm 0.0067}$ |
| TUNEUP w/o syn-tails | $0.0829_{\pm 0.0020}$ | $0.0603_{\pm 0.0017}$ | $0.0184_{\pm 0.0026}$ | $0.0988_{\pm 0.0046}$ | $0.0824_{\pm 0.0021}$ | $0.0710_{\pm 0.0031}$ | $0.1036_{\pm 0.0024}$ | $0.0738_{\pm 0.0021}$ | $0.0273_{\pm 0.0041}$ |
| **TUNEUP (ours)** | $\mathbf{0.1247}_{\pm 0.0032}$ | $\mathbf{0.1092}_{\pm 0.0039}$ | $\mathbf{0.0715}_{\pm 0.0041}$ | $\mathbf{0.1513}_{\pm 0.0070}$ | $\mathbf{0.1328}_{\pm 0.0043}$ | $\mathbf{0.1045}_{\pm 0.0074}$ | $\mathbf{0.1593}_{\pm 0.0041}$ | $\mathbf{0.1252}_{\pm 0.0044}$ | $\mathbf{0.0594}_{\pm 0.0046}$ |
| Rel. gain over base | +50.3% | +80.1% | +285.1% | +52.8% | +59.1% | +47.5% | +56.0% | +73.3% | +147.5% |

Table 11: Cold-start inductive link prediction performance with GCN. The evaluation metric is recall@50. The larger the edge removal ratio is, the more cold-start the prediction task becomes.

| Method | flickr | | | ppi | | | arxiv | | |
|---|---|---|---|---|---|---|---|---|---|
| | Edge removal ratio | | | Edge removal ratio | | | Edge removal ratio | | |
| | 30% | 60% | 90% | 30% | 60% | 90% | 30% | 60% | 90% |
| Base | $0.1144_{\pm 0.0023}$ | $0.0882_{\pm 0.0022}$ | $0.0261_{\pm 0.0012}$ | $0.1083_{\pm 0.0030}$ | $0.0909_{\pm 0.0037}$ | $0.0381_{\pm 0.0031}$ | $0.1717_{\pm 0.0017}$ | $0.1192_{\pm 0.0019}$ | $0.0376_{\pm 0.0015}$ |
| DropEdge | $0.1241_{\pm 0.0027}$ | $0.1006_{\pm 0.0020}$ | $0.0604_{\pm 0.0047}$ | $0.1610_{\pm 0.0067}$ | $\mathbf{0.1364}_{\pm 0.0066}$ | $\mathbf{0.0765}_{\pm 0.0037}$ | $0.1822_{\pm 0.0011}$ | $0.1346_{\pm 0.0010}$ | $0.0597_{\pm 0.0018}$ |
| LocalAug | $0.1222_{\pm 0.0026}$ | $0.0936_{\pm 0.0030}$ | $0.0282_{\pm 0.0024}$ | $0.1151_{\pm 0.0027}$ | $0.0920_{\pm 0.0046}$ | $0.0400_{\pm 0.0044}$ | $0.1772_{\pm 0.0011}$ | $0.1216_{\pm 0.0015}$ | $0.0378_{\pm 0.0012}$ |
| ColdBrew | $0.0937_{\pm 0.0023}$ | $0.0686_{\pm 0.0035}$ | $0.0483_{\pm 0.0028}$ | $0.0388_{\pm 0.0265}$ | $0.0376_{\pm 0.0223}$ | $0.0328_{\pm 0.0142}$ | $0.1399_{\pm 0.0015}$ | $0.0927_{\pm 0.0015}$ | $0.0415_{\pm 0.0018}$ |
| Tail-GNN | $0.1059_{\pm 0.0015}$ | $0.0830_{\pm 0.0030}$ | $0.0541_{\pm 0.0050}$ | $0.1189_{\pm 0.0071}$ | $0.0971_{\pm 0.0053}$ | $0.0560_{\pm 0.0018}$ | $0.1001_{\pm 0.0009}$ | $0.0811_{\pm 0.0014}$ | $0.0468_{\pm 0.0016}$ |
| RAWLS-GCN | $0.0437_{\pm 0.0020}$ | $0.0430_{\pm 0.0017}$ | $0.0421_{\pm 0.0019}$ | $0.0575_{\pm 0.0031}$ | $0.0453_{\pm 0.0010}$ | $0.0394_{\pm 0.0033}$ | $0.0623_{\pm 0.0005}$ | $0.0408_{\pm 0.0014}$ | $0.0239_{\pm 0.0013}$ |
| TUNEUP w/o curriculum | $0.1272_{\pm 0.0022}$ | $0.1048_{\pm 0.0016}$ | $\mathbf{0.0661}_{\pm 0.0021}$ | $0.1437_{\pm 0.0046}$ | $0.1241_{\pm 0.0039}$ | $\mathbf{0.0754}_{\pm 0.0053}$ | $0.1852_{\pm 0.0015}$ | $0.1356_{\pm 0.0017}$ | $0.0557_{\pm 0.0023}$ |
| TUNEUP w/o syn-tails | $0.1166_{\pm 0.0021}$ | $0.0883_{\pm 0.0005}$ | $0.0284_{\pm 0.0010}$ | $0.1110_{\pm 0.0078}$ | $0.0916_{\pm 0.0053}$ | $0.0347_{\pm 0.0036}$ | $0.1737_{\pm 0.0008}$ | $0.1195_{\pm 0.0014}$ | $0.0368_{\pm 0.0010}$ |
| **TUNEUP (ours)** | $\mathbf{0.1308}_{\pm 0.0021}$ | $\mathbf{0.1080}_{\pm 0.0027}$ | $\mathbf{0.0661}_{\pm 0.0028}$ | $\mathbf{0.1714}_{\pm 0.0051}$ | $\mathbf{0.1430}_{\pm 0.0073}$ | $\mathbf{0.0786}_{\pm 0.0090}$ | $\mathbf{0.1933}_{\pm 0.0018}$ | $\mathbf{0.1440}_{\pm 0.0017}$ | $\mathbf{0.0627}_{\pm 0.0015}$ |
| Relative gain over base | +14.4% | +22.5% | +153.2% | +58.2% | +57.4% | +106.4% | +12.6% | +20.9% | +67.0% |

