# OpenReview forum: "TuneUp: A Training Strategy for Improving Generalization of Graph Neural Networks"
_ICLR.cc/2023/Conference — Submitted to ICLR 2023_

### Official Review · Reviewer_1BhX · 2022-10-20

**Confidence:** 4
**Correctness:** 2
**Technical Novelty And Significance:** 3
**Empirical Novelty And Significance:** 3
**Recommendation:** 3

**Clarity, Quality, Novelty And Reproducibility:**

This paper is built on three suspicious assumptions, please refer to “weakness” above.

**Strength And Weaknesses:**

Strength:
1. The authors performed three different types of GNN applications, all with improved results.
2. It is significant to improve the performance of the tail node.

Weakness:

The author's approach is based on three very strong assumptions, which I disagree with, as follows:
1. The first one is "Tail nodes can be synthesized by dropping edges from head nodes".
Why the DropEdge can synthesize the tail node from the head node?
I think in most cases， dropping edges from head nodes would give synthetic tail nodes that have different characteristics from real tail nodes
The authors did not prove this experimentally or theoretically, so I don't think this assumption is convincing enough.

2. The second one is "Target supervision on head nodes can be reused for synthetic tail nodes".
The author points out that the degree of a node is not related to its properties, which I think is incorrect.
In most cases, the degree of a node is strongly related to its own properties.
For example, in a social network, celebrities will have many people to follow, thus resulting in a high degree.
I think the authors should also give explanations from an experimental or theoretical point of view.

3. The third one is "Edge information helps for GNNs to make prediction".
In many cases, more edges are not always better.
Real-life graph data has many heterogeneous connections, and these edges can lead to performance degradation.


**Summary Of The Paper:**

This work proposes a curriculum learning strategy, named TUNEUP for GNN, which aims to improve the performance of GNNs at the tail nodes. TUNEUP first trains a GNN that produces good performance on head nodes, then synthesizes tail nodes by dropping the edges of head nodes, and then fine-tunes on these synthesized tail nodes. The authors conduct three types of experiments: semi-supervised node classification, link prediction and recommender systems. Extensive experiments demonstrate the effectiveness of this method.

**Summary Of The Review:**

I think the authors should give theoretical or experimental explanations and clarifications for these three assumptions.

---

> ### Author Response · Authors · 2022-11-15
> **Response**
>
> We thank the reviewer for the time and comments. Our strong empirical results across different realistic datasets suggest that the assumptions hold (at least approximately) in practice. Given our empirical results on real datasets, we are a little bit confused by the reviewer’s judgment on questioning our assumptions. Below is our response to each individual concern.
>
> ------
>
> **Q:** Concern on the first assumption (tail nodes can be synthesized by removing edges from head nodes)
>
> **A:** We proved empirically that training with DropEdge gives a superior performance on real tail nodes. This suggests that real tail nodes are well-simulated by dropping edges from head nodes.
>
> ------
>
> **Q:** Concern on the second assumption (target supervision on head nodes can be reused)
>
> **A:** In many OGB benchmarks, we aim to predict class labels of nodes, which is irrelevant to the node degrees. Of course, if the goal is to predict the popularity of the nodes, which is highly related to the node degrees, then the label cannot be re-used.
>
> ------
>
> **Q:** Concern on the third assumption (edges are helpful for GNNs to make accurate predictions):
>
> **A:** There is certainly some noise in edges, but we assume more edges are overall helpful for GNNs to make accurate predictions. This is verified in Figure 1, where GNNs’ performance becomes higher as node degrees are higher.

---

### Official Review · Reviewer_Yj9L · 2022-10-24

**Confidence:** 4
**Correctness:** 2
**Technical Novelty And Significance:** 2
**Empirical Novelty And Significance:** Not applicable
**Recommendation:** 3

**Clarity, Quality, Novelty And Reproducibility:**

* **Clarity:** The presentation is clear.
* **Novelty:** The nolvety is weak since the employed DropEdge, Pseudo Label and Curricum Learning method are well explored in previous works. See the 1st weakness in the previous section.
* **Reproducibility:** The reproducibility is in concern due to the abnormal experimental setting. See the 3rd weakness in the previous section.

**Strength And Weaknesses:**

**Strengths:**

* The paper is clearly written.
* The studied problem of generalization to tail nodes is important for recommender systems and social network analysis.
* The proposed model is tested on extensive benchmarks from various domains.



**Weakness**:
-  The technical novelty is weak.
   1. The methods of DropEdge and Pseudo Labeling are well explored in previous works, as already cited in this paper.
   2. The proposed TuneUp method is a straightforward adaptation of the "from easy to difficulty" idea from Curriculum Learning.

- The experimental results are insignificant.
   1. In Table1, *TuneUp w/o pseudo-labels* underperforms *DropEdge*. The result shows that the proposed two-stage training method is no better than a single-stage training with augmentation. Although using pseudo-labels can improve performance, pseudo labeling is proposed by previous works and does not align with the motivation of this paper.
   2. In Table2, TuneUp's improvement over DropEdge is marginal. The result shows that the improvement over the base model is mainly attributed to the DropEdge augmentation rather than the proposed method.

- The experimental settings need more careful treatment.
   1. In sec4.2, the experimental setting of the link prediction task follows that of two papers from a different domain -- recommender system -- instead of link prediction. The authors are encouraged to follow the experimental setting of papers from the same domain for appropriate evaluation.
   2. It is weird to use an ogb**n** (**n** for node) dataset for link prediction evaluation, since the ogb benchmark has multiple ogb**l** (**l** for link) datasets designed specifically for link prediction evaluation.
   3. I also notice that the experimental settings of semi-supervised node classification and recommendation are new and different from existing works. The authors are encouraged to report the performances under the same evaluation protocols of existing works.

**Summary Of The Paper:**

The paper proposes a two-stage curriculum learning strategy -- TuneUp --  to improve GNN's performance on tail nodes of small node degrees. The paper is motivated by the observation that GNNs perform worse on nodes with smaller degrees.

In this first stage, TuneUp conducts standard training without augmentations. In the second stage, TuneUp trains GNNs with DropEdge augmentation. The paper argues that the DropEdge augmentation in the second stage can synthesize tail nodes from head nodes, thus providing more tail nodes for training.

TuneUp is evaluated in three tasks (semi-supervised node classification, link prediction, and recommendation). When the task is semi-supervised node classification, TuneUp also uses pseudo labels in the second stage of training.

**Summary Of The Review:**

The paper presents a two-stage curriculum learning method to improve performance on tail nodes of small degrees. The novelty is weak and the experimental results are insignificant. The experimental settings need more careful treatments. The paper is not ready for publication at the current stage. Hence, I am leaning on the negative side.

---

> ### Author Response · Authors · 2022-11-15
> **Response**
>
> Thank you for your time and comments. Below we address the reviewer’s concerns.
>
> ----
> **Q:** Not following the standardized protocol.
>
> **A:** See our common response (2), where we detail why the conventional standardized protocols are not ideal.
>
> -----
>
> **Q:** The effect of TuneUp is marginal in semi-supervised node classification.
>
> **A:** We respectfully disagree. In semi-supervised learning, our ablation (highlighted in the table below) shows that the combination of pseudo labels and DropEdge (which is TuneUp) is crucial to significantly improve performance. Neither Pure DropEdge nor pure pseudo labels (TuneUp w/o synthetic tail nodes) give as significant improvement as our TuneUp. While DropEdge and pseudo labels are not new, their combination is new, and we clearly demonstrated that the combination is critical. Moreover, while the effect of “finetuning” may indeed be marginal, the pseudo labels cannot be obtained without the base GNN that we obtained in the first stage. In this sense, the two stages are necessary.
>
> | Dataset                         | arxiv        | arxiv      | arxiv            | products     | products   | products         |
> |---------------------------------|--------------|------------|------------------|--------------|------------|------------------|
> | Setting                         | Transductive | Inductive  | Inductive (cold) | Transductive | Inductive  | Inductive (cold) |
> | Base                            | 0.6738       | 0.6689     | 0.4748           | 0.8408       | 0.8424     | 0.7226           |
> | Pure DropEdge                   | 0.6756       | 0.6692     | 0.5446           | 0.8463       | 0.8471     | 0.7710           |
> | Pure pseudo labels              | 0.6785       | 0.6758     | 0.4900           | 0.8434       | 0.8450     | 0.7255           |
> | TuneUp (DropEdge+pseudo labels) | **0.6867**   | **0.6784** | **0.6006**       | **0.8554**   | **0.8564** | **0.8054**       |
>
>
> -------
>
> **Q:** Technical novelty is weak.
>
> **A:** While our method is simple, we believe our findings and results are novel and non-trivial. See our common response (3).
>
> ------
>
> **Q:** The improvement of TuneUp on link prediction is marginal.
>
> **A:** Yes, for link prediction, the major improvement in link prediction is attributed to DropEdge. However, TuneUp is generally is effective *across* three tasks, while the vanilla DropEdge is not. Moreover, the DropEdge is originally developed for node classification, and our work finds that DropEdge is surprisingly effective in link prediction, which is non-trivial and valuable.

---

### Official Review · Reviewer_xBXx · 2022-10-25

**Confidence:** 4
**Correctness:** 3
**Technical Novelty And Significance:** 2
**Empirical Novelty And Significance:** 3
**Recommendation:** 3

**Clarity, Quality, Novelty And Reproducibility:**

Clarity:
- Clear.

Quality:
- Good

Novelty:
- Small

Reproducibility:
- Good

**Strength And Weaknesses:**

Strengths:
- Clear communication, easy to understand
- Discusses the assumptions that enable the method to perform better.
- General method, i.e. applicable in many areas
- Good dataset selection to evaluate empirical performance
- Covers both transductive and inductive settings, as well as cold-start nodes
- Ablation studies done are meaningful and reinforce the points of the paper
- Many other graph augmentation methods are compared against

Weaknesses:
- The GNN architecture used in evaluation is old and a narrow choice. The most standard choice would be an MPNN and then checking with the 3 most common aggregators, which I suspect would have a large impact on how the node degrees affect prediction. So I think the information provided by the empirical evaluation may not be useful to the community as GraphSAGE is simply not the default choice anymore. GCN is even older. The authors should rerun at least the largest dataset in each task (node classification, link prediction, recommender system) with the three MPNN (max, mean, sum aggregators).
- The evaluation does not do enough to convince the reader that this is useful across GNN architectures, for that MPNN with different aggregators are the minimum.
- The deviation in the evaluation protocol for node classification, while sensible, is unfortunate in that it prevents easy comparison for past, concurrent, and future work as it is not standard.
- For link prediction there is no test set, which I find problematic and is not an acceptable protocol for a top machine learning conference.

Minor:
- The standard deviation numbers in a lot of tables (especially Table 2.) are too small to be readable, the authors would be better off reporting them in the appendix and removing them from the main paper. It harms the readability too significantly.

Questions:
- You claim in section 2.4 that links in heterophilieous graphs would necessarily harm predictive performance in node classification. I can't imagine that is true, consider graph colouring, where adjacent nodes needs to have different colours and we are given an initial colouring for some of the nodes. This is a node classification problem in a heterophelic graph, link information is informative, in fact necessary for prediction. Am I missing something or are heterophileous graphs not a good example of the point you are trying make?


**Summary Of The Paper:**

The paper proposes a two-stage curriculum learning process for GNNs to ensure better performane in a variety of tasks. Specifically, the paper argues that low-degree nodes are the hardest to make predictions for and focus on improving this.

**Summary Of The Review:**

The paper is clear and well written. The method is a small technical innovation that seems applicable across a wide variety of tasks. However, the empirical evaluation needs to be improved, in particular the GNN architectures considered as well as using a test set for link prediction.

If these concerns are addressed, I am willing to raise my score.

---

> ### Author Response · Authors · 2022-11-15
> **Response**
>
> Thank you for your time and comments. Below we address the reviewer’s concerns.
>
> -----
> **Q:** Only basic GNN models are tested.
>
> **A:** See our common response (1) and Table 4 of the updated manuscript. We have added 3 more GNN architectures to our experiments.  In total, we now have 5 diverse GNN models, covering mean, re-normalized mean, max, sum, and attention aggregations, over which TuneUp shows strong performance.
>
> -----
> **Q:** Not following the standardized protocol
>
> **A:** See our common response (2), where we detail why the conventional standardized protocols are not ideal.
>
> -----
> **Q:** Lack of validation performance in link prediction.
>
> **A:** Although this is true for the transductive setting, our inductive performance, and cold inductive performance are fully test performances. From Table 2, we see that the TuneUp consistently performs well there, giving us confidence that we did not overfit the validation set. Moreover, there are definitely well-cited top conference papers that report validation performance in the link prediction / recsys [1,2,3].
>
> -----
> **Q:** Why links in heterophyllous graphs would necessarily harm predictive performance in node classification?
>
> **A:** This is because many GNNs have an inductive bias of making smooth predictions over graphs by smoothing the neighboring features [4]. As a result, the edges can harm the performance of many GNNs. We have clarified the point in our updated manuscript.
>
> -----
> [1] https://arxiv.org/abs/1802.09691
>
> [2] https://arxiv.org/abs/1708.05031
>
> [3] https://arxiv.org/abs/2002.02126
>
> [4] https://arxiv.org/abs/1902.07153

---

> > ### Comment · Reviewer_xBXx · 2022-12-01
> > **Testing on the validation set is not a valid machine learning approach**
> >
> > As with most fields, as our understanding grows, we realise that previous evaluation protocols were unwise. Testing on validation sets is simply not acceptable anymore (even if it may have been done in previous papers). Unless this is fixed I simply cannot recommend the paper for acceptance.

---

### Official Review · Reviewer_khNG · 2022-10-25

**Confidence:** 3
**Correctness:** 2
**Technical Novelty And Significance:** 2
**Empirical Novelty And Significance:** 2
**Recommendation:** 3

**Clarity, Quality, Novelty And Reproducibility:**

The writing conveys the ideas of the authors, but the organization of the paper is not very good. There are a few assumptions not well supported by experiments.

**Strength And Weaknesses:**

### Strengths
1. The paper introduces a training strategy, TuneUp, that organically combines multiple data augmentation methods for GNN.
2. The proposed TuneUp strategy is intuitive and performs well on multiple types of prediction tasks.
3. It also considers an interesting cold-start scenario that may be useful for real applications.

### Weaknesses
1. The paper claims the TuneUp strategy can improve the performance of the GNN training. However, it is only tested on the basic GNN models, GraphSage and GCN. It would be interesting to see other GNN architectures that appear in the ogbn leaderboard to show the generality of the method.
2. It would be interesting to see some examples of nodes and targets that are generated from the synthesis stage to give more insights into the method.
3. The method adopts pseudo labels to improve performance. It would be also great to have some label-smoothing baselines.
4. The paper made several assumptions for the augmentation techniques used in the algorithm. It can be more convincing to show some statistics of the graphs in different datasets, such as how many pseudo-labels are correct, and the accuracy for the head nodes vs tail nodes for the base model

**Summary Of The Paper:**

The paper proposes a training strategy for GNN, TuneUp, that applies curriculum learning with node synthesis and label generation. It is evaluated on multiple modalities and outperforms multiple data augmentation baselines.

**Summary Of The Review:**

The paper proposes a training strategy for GNN that combines multiple data augmentation methods together and improves the performance of two basic GNN models. It is not well-written and several assumptions are not well supported.

---

> ### Author Response · Authors · 2022-11-15
> **Response**
>
> Thank you for your time and comments. Below we address the reviewer’s concerns.
>
> ----
>
> **Q:** Only basic GNN models are tested.
>
> **A:** See our common response (1) and Table 4 of the updated manuscript. We have added 3 more GNN architectures to our experiments.  In total, we now have 5 diverse GNN models, covering mean, re-normalized mean, max, sum, and attention aggregations, over which TuneUp shows strong performance.
>
> ----
>
> **Q:** The method adopts pseudo labels to improve performance. It would be also great to have some label-smoothing baselines.
>
> **A:** Our TuneUp allows users to select any kind of pseudo labels (not just those directly generated by the base GNN). For example, one can use some label smoothing techniques (such as C&S by https://arxiv.org/abs/2010.13993) to refine the pseudo labels before the base GNN is finetuned on the pseudo labels. Therefore, label smoothing is complementary to our TuneUp.  We have clarified this in our updated manuscript.
>
> -----
>
> **Q:** Detailed statistics about the model performance
>
> **A:** We have provided all the statistics that the reviewer has raised. “how many pseudo-labels are correct” is essentially the accuracy of the base GNN, and “the accuracy for the head nodes vs tail nodes for the base model” is extensively shown in Figures 1, 2, 3, and 4.
>
> -----
>
> **Q:** It would be interesting to see some examples of nodes and targets that are generated from the synthesis stage to give more insights into the method.
>
> **A:** We can certainly show nodes’ neighborhoods and their targets, but they are not easily human interpretable, unlike images and texts. We would appreciate the reviewer’s concrete suggestions on how to best show examples.

---

### Author Response · Authors · 2022-11-14
**Common response**

We thank all the reviewers for their reviews and constructive feedback. Overall, we are glad that the reviewers like the idea and effectiveness of the TuneUp method. At the same time, the reviewers raise many critical concerns. We have addressed the concerns thoroughly and updated the manuscript accordingly, which significantly strengthened our work. In the common response here, we address three important concerns raised by multiple reviewers: (1) Lack of GNN architectures tested, (2) Not following the standardized experimental protocols, and (3) Lack of technical novelty.

**(1) Lack of GNN architectures tested.**

Thank you for raising this important point. We agree that testing TuneUp on more GNN architectures would strengthen our paper. Therefore, **we have added Table 4 in the updated manuscript**, where we included three additional models, SAGE-max*, SAGE-sum*, and GAT.  **In total, we have five diverse GNN architectures** that cover representative aggregation schemes (i.e., mean, renormalized-mean, max, sum, and attention). From Table 4, we observed (i) **consistent gains across diverse models**, indicating the broad applicability of TuneUp, and (ii) the original SAGE and GCN are still strong models among the 5 GNN architectures.

\* SAGE-max and SAGE-sum are the variants of the original GraphSAGE, where we replaced the mean aggregation with max and sum aggregations, respectively.

**(2) Not following the standard protocols**

While we agree that it is important to follow the standardized protocol as much as we can, we noticed that **the standardized experimental protocols by OGB and recommender systems are not suitable for evaluating TuneUp** because (1) inductive prediction (cold-start) settings are not provided and (2) datasets are heavily pre-processed to eliminate tail nodes (e.g., recommender system benchmarks are processed with the 10-core algorithm to eliminate the cold-start users and items), which is the focus of this work. We, therefore, take their original realistic graph datasets and split them ourselves to create the realistic inductive (cold-start) prediction setting as well as the realistic transductive setting with tail nodes. We will make sure that our protocols can be easily reproduced by follow-up works.
Below, we share more details about why it is better/fine to use our own protocols than the conventional protocol.
- OGB’s node classification/link prediction datasets do not provide inductive settings, which is the key in our evaluation.
- OGB’s node classification (e.g., ogbn-arxiv) has 60% of nodes labeled, which deviates from the semi-supervised node classification setting.
- The evaluation of OGB’s link prediction is not rigorous in the sense that positive edges are ranked against randomly-selected negative edges (without even ensuring the end-nodes are shared). In contrast, we fixed the source node and evaluate the target positive node against all negative nodes. Our evaluation is more practically meaningful and realistic [2].
We initially followed the conventional recsys settings (80%/20% train/val split) but then realized that the average/median training edges per user were very high (e.g., 45/27 for the amazon-book), which clearly neglects real-world scenarios due to the heavy 10-core pre-processing. When we tried the TuneUp strategy on the conventional high-degree setting, it is not surprising to see TuneUp did not give much improvement over the base GNN.
- Moreover, on the new split and settings, we have provided a sufficient number of baselines to make sure our TuneUp is compared against meaningfully strong baselines.
- We will make sure to open-source code / split so that it is easy to follow up with our work to study the tail node prediction settings.
- Finally, in order not to confuse the readers, in the updated paper, we have removed the “ogbn-” header from our dataset name to emphasize that we did not follow OGB’s standardized protocol.

**(3) Lack of technical novelty**

As we have emphasized in the paper, **our novelty is focusing on the training strategy of GNNs**. As our work is the first to investigate the training strategy, we opted for simplicity without inventing an overly-complex strategy. Despite the simplicity, TuneUp outperforms many complicated/specialized existing methods across different task types and also improves the performance on head nodes (beyond the tail nodes). The experimental results are surprising and highly non-trivial.

[1] https://arxiv.org/abs/1902.07153

[2] https://dl.acm.org/doi/pdf/10.1145/3394486.3403226 (KDD best paper 2020)

---

### Decision · Program_Chairs · 2023-01-20

**Decision:**

Reject

**Justification For Why Not Higher Score:**

The reviewer raised concerns including need for testing on extensive set of GNNs, using synthetic examples of nodes and targets to give more insights of algorithms, need for some label-smoothing baselines, need for test set for link prediction, limited novelty. The authors addressed some of the commented issues during response period. However, overall, there still seems to be a lack of enough enthusiasm among the reviewers after the author responses.

**Justification For Why Not Lower Score:**

N/A

**Metareview: Summary, Strengths And Weaknesses:**

The paper proposes a training strategy for GNN, TuneUp, a two-stage curriculum learning strategy that applies curriculum learning with node synthesis and label generation. In this first stage, TuneUp conducts standard training without augmentations. In the second stage, TuneUp trains GNNs with DropEdge augmentation. Specifically, the paper argues that low-degree nodes are the hardest to make predictions for and focus on improving this. TuneUp is evaluated in three tasks (semi-supervised node classification, link prediction, and recommender systems). When the task is semi-supervised node classification, TuneUp also uses pseudo labels in the second stage of training. The reviewer raised concerns including need for testing on extensive set of GNNs, using synthetic examples of nodes and targets to give more insights of algorithms, need for some label-smoothing baselines, need for test set for link prediction, limited novelty. The authors addressed some of the commented issues during response period. However, overall, there still seems to be a lack of enough enthusiasm among the reviewers after the author responses.